# EgoBridge: Domain Adaptation for Generalizable Imitation from Egocentric Human Data

**Ryan Punamiya[1]**      **Dhruv Patel[1]**      **Patcharapong Aphiwetsa[1]**      **Pranav Kuppili[1]**
**Lawrence Y. Zhu[1]**      **Simar Kareer [1*]**      **Judy Hoffman[1*]**      **Danfei Xu[1*]**
[1]Georgia Institute of Technology      [*]Equal advising

rpunamiya6@gatech.edu

## Abstract

Egocentric human experience data presents a vast resource for scaling up end-to-end imitation learning for robotic manipulation. However, significant domain gaps in visual appearance, sensor modalities, and kinematics between human and robot impede knowledge transfer. This paper presents EgoBridge, a unified co-training framework that explicitly aligns the policy latent spaces between human and robot data using domain adaptation. Through a measure of discrepancy on the joint policy latent features and actions based on Optimal Transport (OT), we learn observation representations that not only align between the human and robot domain but also preserve the action-relevant information critical for policy learning. EgoBridge achieves a significant absolute policy success rate improvement by 44% over human-augmented cross-embodiment baselines in three real-world single-arm and bimanual manipulation tasks. EgoBridge also generalizes to new objects, scenes, and tasks seen *only* in human data, where baselines fail entirely. Videos and additional information can be found at https://ego-bridge.github.io/

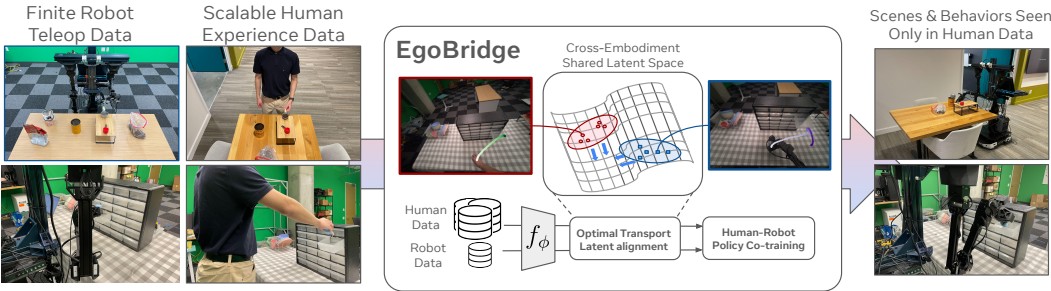

Figure 1: EgoBridge enables rich knowledge transfer from human to robot, based on our key hypothesis: aligned latent representations yield stronger transfer. Our algorithm, which adapts optimal transport, aligns behaviors which are similar across embodiments. This enables EgoBridge to generalize to objects, scenes and even motions demonstrated only in human data.

## 1 Introduction

Supervised imitation learning methods such as behavior cloning have emerged as a promising path to scaling robot performance across diverse objects, tasks, and environments. However, while large-scale models in vision and language have achieved remarkable generalization through Internet-sourced data, replicating this success in robotics remains challenging due to the labor-intensive nature of collecting teleoperated demonstrations. Deploying physical robots to many new environments to collect data with enough coverage and diversity is economically and practically intractable.

39th Conference on Neural Information Processing Systems (NeurIPS 2025).

In this work, we aim to enable robots to learn from egocentric recordings of natural human behavior, collected by increasingly ubiquitous wearable devices (e.g., XR devices and smart glasses). Without a robot in the loop, such data is cheap and scalable to collect and captures natural human interactions with the world. More importantly, it reflects the *embodied human experience*, as it contains both observations (e.g., egocentric RGB images) and actions (e.g., hand motions). Unlike unstructured data sources such as Internet videos, the rich embodied information allows us to treat human data and robot data as equal parts in a continuous spectrum of demonstration data and potentially learn from both with a unified learning framework.

However, the multitudes of *domain gaps* between human and robot pose significant challenges in designing such a framework. Human bodies and robots have different visual appearances. Even within a shared action space, kinematic differences can lead to behavior distribution shifts. Robots also have additional sensing modalities such as wrist cameras that are often missing from embodied human data. While recent works such as EgoMimic [1] have attempted to bridge the embodiment gaps with techniques such as visual masking, data normalization, and motion retargeting, such domain gaps still largely remain. More broadly, simply co-training from cross-domain data does not automatically yield effective knowledge transfer, as suggested by recent studies [2]. Such challenges prevent policies from scaling their performance primarily with human data.

We formalize the human-robot cross-embodiment learning problem as a *domain adaptation problem*, where human and robot data represent two labeled distributions with significant *covariate shifts* in observations due to embodiment gaps. Standard domain adaptation approaches often rely on global distribution alignment techniques such as adversarial training [3] and maximum mean discrepancy minimization [4]. However, they primarily address high-level tasks such as image classification and fail to preserve detailed action-relevant information—a critical requirement for robot learning where actions and observations are temporally correlated under compounding covariate shift.

To address these challenges, we propose **EgoBridge**, a novel domain adaptation approach that uses Optimal Transport (OT) to align latent representations from human and robot domains as part of the policy co-training objective. Unlike conventional domain alignment methods, our OT formulation explicitly exploits the inherent relationship between motion similarities in human and robot domains to form *pseudo-pairs* as supervision for the adaptation process. Concretely, we use the dynamic time warping (DTW) distance among human and robot motion trajectories to shape the OT ground cost. This encourages the transport map to find a minimal-cost coupling between human and robot data exhibiting similar behaviors. As such, EgoBridge aligns policy representations across domains via a differentiable OT loss (Sinkhorn distance), while preserving action-relevant information for policy learning. Importantly, we show that EgoBridge learns a shared latent representation that *generalizes* beyond the paired data. This enables the policy to learn behaviors observed *only within the human dataset*, effectively enabling the policy to scale primarily with human data.

We evaluate **EgoBridge** on both a reproducible simulation benchmark task and three challenging real-world manipulation tasks. Our results show that **EgoBridge** consistently improves policy success rates compared to human-augmented cross-embodiment baselines, for up to 44% absolute success rate improvement, and effectively transfers behaviors from diverse human demonstrations to robotic execution in tasks requiring spatial, visual, and task generalization.

## 2   Related Work

**Supervised Imitation Learning.** Supervised Imitation Learning (SIL), notably Behavior Cloning, leverages expert demonstrations for policy learning and has achieved significant success in robotics, particularly with large-scale datasets [5, 6, 7, 8, 9, 10, 11, 12, 13, 14]. State-of-the-art Vision-Language-Action (VLA) models [10, 8, 12, 11] integrate Vision-Language Models (VLMs) with action decoders, enhancing generalization by incorporating semantic understanding from internet-scale data. Despite these advances, even top-performing models like Pi 0.5 [11] require extensive labeled robot manipulation data for robust physical world interaction and broad capabilities. Given the intractability of scaling robot teleoperation data, alternative sources such as more accessible human data are being explored. Consequently, our work leverages scalable human demonstrations alongside in-domain robot data to improve learning outcomes.

**Learning from Human Data.** Human data presents two main opportunities for robot learning: abundant unlabeled online videos and curated, labeled demonstrations [15, 16, 1]. Unlabeled web

videos, though plentiful, require pseudo-labeling of actions via inverse dynamics models [17] or point tracking [18, 19, 20] for policy training, forming a basis for some foundation models [12], yet often still necessitating in-domain robot data. Alternatively, labeled human demonstrations can be co-trained with robot data as distinct embodiments [1, 21, 22], enhancing robustness and scene understanding. However, generalizing to novel behaviors observed only in human data remains challenging. To address this limitation, we propose a novel learning framework for jointly aligning observation-action spaces across human and robot embodiments to improve generalization.

**Domain Adaptation and Optimal Transport.** Domain Adaptation (DA) aims to reduce reliance on target-specific data by leveraging labeled source domain data to bridge distribution gaps and improve performance on unlabeled target domains. In cross-embodiment learning, DA has been applied for shared dynamics modeling [23], unsupervised reward modeling [24], and high-level planning [25]. However, many DA methods primarily focus on global distribution alignment, which can neglect fine-grained action information crucial for transfer across robot embodiments. To address this, computer vision research introduced Optimal Transport (OT) as a loss function for DA to align both local and global distributions [26, 27, 28]. Building on these insights, we propose an action-aware DA approach using OT to learn shared representations across embodiments, thereby improving observation and behavior generalization.

## 3 Preliminaries and Problem Statement

### 3.1 Optimal Transport

**Optimal Transport for Domain Adaptation.** Optimal Transport (OT) offers a principled framework for comparing probability distributions by considering the geometry of their sample spaces. Given two distributions, $\mu_S$ (source) and $\mu_T$ (target), over a common metric space $\mathcal{X}$, and a cost function $\mathcal{C}(x^S, x^T)$ measuring the effort to move mass from $x^S \in \mathcal{X}$ to $x^T \in \mathcal{X}$, OT finds a probabilistic coupling $\gamma \in \mathcal{P}(\mathcal{X} \times \mathcal{X})$ that minimizes the expected transport cost:

$$\gamma^* = \arg\min_{\gamma \in \Pi(\mu_S, \mu_T)} \mathbb{E}_{(x^S, x^T) \sim \gamma}[\mathcal{C}(x^S, x^T)],$$

where $\Pi(\mu_S, \mu_T)$ is the set of all joint distributions whose marginals are $\mu_S$ and $\mu_T$. For discrete empirical distributions from $N_S$ source samples $\{x_i^S\}$ and $N_T$ target samples $\{x_j^T\}$, the cost matrix is $C_{ij} = \mathcal{C}(x_i^S, x_j^T)$, and the total cost is $\langle \gamma, C \rangle_F = \sum_{i,j} \gamma_{ij} C_{ij}$.

**Differentiable Optimal Transport as a Loss Function.** When used as a cost function to align representations, the standard OT problem is often regularized. The Sinkhorn algorithm [29] introduces an entropic regularization term to the OT objective, yielding a differentiable approximation $T_\epsilon^*$ to the optimal transport plan:

$$T_\epsilon^* = \arg\min_{T \in \Pi(\mu_S, \mu_T)} \mathbb{E}_{(x^S, x^T) \sim T}[\mathcal{C}(x^S, x^T)] - \epsilon H(T),$$

where $\epsilon > 0$ is the regularization strength and $H(T)$ is the entropy of the coupling. This regularization makes the problem strictly convex and efficiently solvable. The resulting regularized optimal transport cost, $\sum_{i,j}(T_\epsilon^*)_{ij} C_{ij}$, is differentiable with respect to the cost matrix $C$. This allows OT to serve as a loss function within deep learning frameworks, enabling the learning of feature encoders that map inputs to a space $\mathcal{X}$ where their distributions are aligned by minimizing this transport cost.

### 3.2 Human and Robot Data Sources

We consider egocentric human data ($\mathcal{D}_H$) and teleoperated robot data ($\mathcal{D}_R$). $\mathcal{D}_H = \{(o_t^H, a_t^H)\}_{t=1}^{N_H}$ consists of $N_H$ egocentric human demonstrations, where $o_t^H \in \mathcal{O}^H$ are observations from wearable sensors (e.g., head-mounted cameras) and $a_t^H \in \mathcal{A}$ are human actions in a common action space (e.g., robot end-effector and human hand poses). This data is abundant and captures natural, diverse behaviors. Conversely, $\mathcal{D}_R = \{(o_t^R, a_t^R)\}_{t=1}^{N_R}$ comprises $N_R$ robot experiences, typically from teleoperation, with $o_t^R \in \mathcal{O}^R$ being robot sensor observations (e.g., ego-centric/wrist cameras, joint states) and $a_t^R \in \mathcal{A}$ the robot actions. This data is often scarce. We describe how each data source is captured and processed in more detail in Sec. 4.3. We assume actions are in trajectory chunks, which is shown to improve prediction temporal consistency of the trained policies [5, 6].

### 3.3 Cross-Embodiment Imitation Learning: Challenges and Objectives

Our primary goal is to effectively learn from both limited robot demonstrations ($D_R$) and more abundant, diverse egocentric human demonstrations ($D_H$). We train a feature encoder $f_\phi : \mathcal{O}^H \cup \mathcal{O}^R \to \mathcal{Z}$ to project observations from both human ($\mathcal{O}^H$) and robot ($\mathcal{O}^R$) into the shared latent space $\mathcal{Z}$. We jointly train a policy $\pi_\theta$ that maps these learned latent representations $z \in \mathcal{Z}$ to actions $a \in \mathcal{A}$.

**Cross-Embodiment Co-Training.** A popular approach [1, 21] involves training the policy end-to-end using a standard Behavior Cloning (BC) loss on the aggregated dataset:

$$\mathcal{L}_{\text{BC-cotrain}}(\phi, \theta) = \mathbb{E}_{(o,a) \sim D_H \cup D_R} [\mathcal{L}_{\text{BC}}(\pi_\theta(f_\phi(o)), a)],$$

To effectively learn from both data sources, a critical assumption is that a shared latent space $\mathcal{Z}$ would naturally emerge where the mapping from latent states to actions is domain-invariant, resulting in $P_R(a|f_\phi(o_R)) \approx P_H(a|f_\phi(o_H))$ for observations $o_R$ and $o_H$ from aligned underlying states.

**Challenge: Observation Covariate Shift.** However, we argue that, without explicit mitigation, the induced marginal distributions over these latents, $\mu_H = P(f_\phi(\mathcal{O}^H))$ and $\mu_R = P(f_\phi(\mathcal{O}^R))$, will exhibit a significant *covariate shift* ($\mu_H \neq \mu_R$). This shift arises from inherent domain gaps in observations (e.g., differing visual appearances, viewpoints, sensor modalities like robot wrist cameras absent in human setups) and embodiment kinematics. We also empirically show that such co-trained representations often form disjoint latent clusters (Sec. 5.3). This covariate shift in the marginal latent distributions undermines the foundational assumption of consistent conditional action distributions across domains, thereby limiting effective knowledge transfer from human to robot.

**Goal: Generalizable Cross-Embodiment Transfer.** This motivates our method that aims at *joint domain adaptation* (Sec. 4.1), where we explicitly seek to align the latent representations from human and robot data while preserving action-relevant information. Successfully addressing this latent misalignment should enable two crucial levels of generalization: First, for tasks present in both $\mathcal{D}_H$ and $\mathcal{D}_R$, the system must achieve **observation generalization**. This involves effectively bridging visual and sensor gaps, which include appearance changes that do not affect behaviour. Second, and more ambitiously, the system should enable **behavior generalization** *(Beh. Gen.)* allowing the robot to perform tasks or handle novel situations (e.g. task variations) observed *only* in $\mathcal{D}_H$. This requires the learned encoder $f_\phi$ to generalize beyond scenarios with paired human and robot data which require motion information to be transferred, such as spatial variations in goal pose.

## 4 EgoBridge

EgoBridge is a co-training framework designed to effectively imitate embodied human demonstrations and robot demonstrations. It explicitly addresses the domain gap between human and robot experiences through an Optimal Transport (OT)-based domain adaptation mechanism integrated into the policy learning process. The core of EgoBridge lies in aligning the *joint distributions of latent policy features and corresponding actions* across the human and robot domains. The following sections detail this joint domain adaptation formulation (Sec. 4.1), the design of its OT cost function (Sec. 4.2), and the overall training process and system details (Sec. 4.3).

### 4.1 Joint Domain Adaptation via Optimal Transport

To address the latent covariate shift (Sec. 3.3) and generalizable cross-embodiment transfer, EgoBridge builds on Optimal Transport (OT, Sec. 3.1) to directly shape the shared feature encoder $f_\phi$. Unlike standard domain adaptation techniques [30] that often aligns only the marginals $P(f_\phi(O))$, which can discard action-relevant information, EgoBridge optimizes $f_\phi$ to align the joint distributions of its output latent features and their corresponding actions, i.e., $P(f_\phi(O), A)$.

Given mini-batches of human data $\{(o_i^H, a_i^H)\}_{i=1}^{N_H}$ and robot data $\{(o_j^R, a_j^R)\}_{j=1}^{N_R}$, where $a$ represents a temporally-extended action trajectory, we define an OT-based loss to guide the learning of $f_\phi$. The differentiable Sinkhorn OT formulation [29] allows us to compute a loss based on the alignment of the empirical distributions of $(f_\phi(o^H), a^H)$ and $(f_\phi(o^R), a^R)$:

$$\mathcal{L}_{\text{OT-joint}}(\phi) = \sum_{i,j} (T_\epsilon^*)_{ij} \cdot \mathcal{C} \left( (f_\phi(o_i^H), a_i^H), (f_\phi(o_j^R), a_j^R) \right).$$

Here, $(T_\epsilon^*)_{ij}$ is the optimal transport plan coupling the $i$-th human and robot (latent, action) pairs. The cost function $\mathcal{C}(\cdot, \cdot)$ measures the dissimilarity between these joint entities. Its design is crucial for capturing meaningful behavioral similarities across domains, which we detail in Sec. 4.2.

Minimizing $\mathcal{L}_{\text{OT-joint}}(\phi)$ directly influences the parameters $\phi$ of the encoder $f_\phi$. The gradients from this loss encourage $f_\phi$ to produce latent features $f_\phi(o_i^H)$ and $f_\phi(o_j^R)$ that minimize the transport cost required to align them, especially when their associated actions $a_i^H$ and $a_j^R$ are behaviorally similar (as determined by $\mathcal{C}$). At each step a transport plan is computed which influences the feature encoder to couple the action pairs This iterative process shapes the latent space $\mathcal{Z}$ to be domain-invariant with respect to the joint observation-action manifold.

## 4.2 Designing OT Cost Function for Action-Aware Joint Adaptation

Our joint OT formulation (Section 4.1) relies on a cost function $\mathcal{C}((z^H, a^H), (z^R, a^R))$ to measure the dissimilarity between joint human and robot latent feature-action pairs. A critical challenge is designing this cost to be robust to inherent domain differences. Specifically, we aim to account for *temporal misalignments*, where human and robot often execute the same task at of different speed, e.g., humans might be 2-3 times faster than teleoperated robots, and *kinematic variations*, where even within a shared SE(3) end-effector action space and hand-eye alignment through an egocentric coordinate frame (Sec. 4.3), minor kinematic differences exist.

**Dynamic Time Warping.** To identify behaviorally similar action sequences while accounting for these differences, we propose to leverage Dynamic Time Warping (DTW) to guide the OT alignment. DTW [31] has been effective in prior work to compare time series data and trajectories. Formally, given two action sequences $\mathbf{a}^H = (a_1^H, \ldots, a_T^H)$ from human data and $\mathbf{a}^R = (a_1^R, \ldots, a_T^R)$ from robot data of identical length $T$, DTW finds an alignment path $\pi \subseteq \{1, \ldots, T\} \times \{1, \ldots, T\}$ that minimizes the cumulative distance:

$$\text{DTW}(\mathbf{a}^H, \mathbf{a}^R) = \min_{\pi \in \mathcal{A}(T)} \sum_{(i,j) \in \pi} \|a_i^H - a_j^R\|^2$$

where $\mathcal{A}(T)$ is the set of admissible monotonic alignments constrained to start at $(1, 1)$ and end at $(T, T)$, while allowing small local shifts to account for temporal variations.

**Soft Supervision**. With DTW, we can identify highly correlated samples from both domains. However, directly utilizing the DTW cost is noisy and instead is a much stronger measure of relative pairing between human and robot samples. As such the DTW cost can be used to *pseudo-pair* $D_H$ and $D_R$. On a mini-batch of size $B$ of sampled ground-truth state-action pairs from $D_S$ and $D_T$, we form a DTW cost matrix $A \in R^{B \times B}$. Here, $A_{i,j} = \text{DTW}(a_i^S, a_j^T)$. The row-wise minimum cost gives us the most behaviorally similar human pseudo-pair for each robot sample : $i^*(j) = \arg \min_i A_{ij}$.

Given the standard OT Euclidean distance cost $D_{ij} = \|f_\phi(o_i^H) - f_\phi(o_j^R)\|^2$, we define the joint cost $\tilde{\mathcal{C}}((f_\phi(o_i^H), a_i^H), (f_\phi(o_j^R), a_j^R))$ for $L_{OT}$ as:

$$\tilde{C}_{ij} = \begin{cases} D_{ij} \cdot \lambda & \text{if } i = i^*(j) \\ D_{ij} & \text{otherwise} \end{cases}$$

where $0 < \lambda \ll 1$ is a small scalar. This cost function strongly incentivizes OT to match robot samples with their behaviorally closest human pseudo-pair (identified by DTW) by significantly reducing the cost for these pairs. For all other pairs, the cost is simply the distance in the latent feature space. This soft supervision from DTW guides the latent space alignment towards behaviorally relevant correspondences across embodiments.

## 4.3 Putting it all together: EgoBridge

With all the ingredients for joint distribution adaptation using OT with joint policy co-training, we present EgoBridge as a unified cross-embodiment imitation learning algorithm and describe its corresponding robot learning system and policy architecture.

**Policy Co-Training with Joint Adaptation.** EgoBridge jointly optimizes the feature encoder $f_\phi$ and $\pi_\theta$, with the joint OT loss applied on the feature encoder and the BC co-training loss applied end-to-end through both components: $\mathcal{L}_{\text{Total}} = \mathcal{L}_{\text{BC-cotrain}}(\phi, \theta) + \alpha \mathcal{L}_{\text{OT-joint}}$, with tunable weight $\alpha$. Detail of the algorithm and hyperparameter choices are described in Appendix.

**Egocentric Human Data.** We largely follow EgoMimic [1] and leverage a wearable smart glass platform Meta Project Aria [32] as our main data collection platform. The platform allows us to collect *exteroceptive*, egocentric first person POV RGB images ($I_{ego}^H$), and *proprioceptive* data ($q^H$), cartesian pose for both arms $\in SE(3) \times SE(3)$. We take inspiration from EgoMimic to construct stable reference frames to form action sequences of cartesian pose ($a^H$) in the egocentric camera frame.

**Teleoperated Robot Data.** We base our robot platform on the open-source Eve robot [1]. In particular, we leverage Aria glasses as the main egocentric perception sensor for the robot and mount it in a way that emulates the hand-eye configuration of a human adult ($I_{ego}^R$). This effectively mitigates the human-robot camera device gap, allowing us to specif-

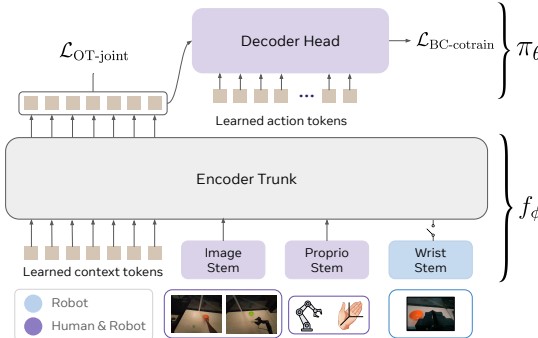

Figure 2: EgoBridge policy co-training with joint adaptation. The encoder $f_\phi$ consists of modality-specific input stems and the encoder trunk, while the policy $\pi_\theta$ consists of a shared multi-block transformer decoder. $\mathcal{L}_{\text{OT-joint}}$ optimizes the encoder while $\mathcal{L}_{\text{BC-cotrain}}$ optimizes the entire network.

ically study the appearance, kinematic and behaviour gaps. The robot additionally provides RGB streams from its two RealSense D405 wrist cameras, $I_{wrist}^R$. The actions consist of a sequence of corresponding future end-effector poses, $a^R \in SE(3) \times SE(3)$.

**Shared Policy Architecture.** Inspired by recent cross-embodiment policy learning [33] and DETR-style architectures [34], our policy employs a shared transformer encoder "trunk" ($f_\phi$) and a shared transformer policy decoder "head" ($\pi_\theta$) (Fig. 2). We perform embodiment-specific gaussian normalization to the proprioception and actions. The encoder $f_\phi$ begins with *stems*—shallow networks that tokenize raw observations; notably, a *shared* vision stem processes main egocentric RGB images ($I_{ego}$) from both human and robot to enforce visual alignment, while separate stems handle robot wrist camera inputs ($I_{wrist}$). The subsequent multi-layer encoder trunk processes these concatenated tokens, along with $M$ prepended learnable context tokens upon which the OT loss is applied. The multi-block decoder head then generates actions by attending to this encoded context, utilizing $T$ learnable action tokens and injecting context through alternating self and cross-attention blocks.

## 5 Experiments

In this section, we aim to validate three core hypotheses. **H1**: EgoBridge improves co-training performance for scenarios present in both human and robot data. **H2**: EgoBridge enables generalization to scenarios only seen in human data. **H3**: EgoBridge learns a shared latent space where human and robot data are aligned in task-relevant manners. We validate the hypotheses through a standard simulation benchmark task (Sec. 5.1) and three complex real-world manipulation tasks (Sec. 5.2).

### 5.1 Simulation Evaluation

To facilitate reproducible study and eliminate the confounding factors in real robot systems, we study a well-explored planar pushing task [6], where the goal is to push a T-shaped object to a desired goal location. We emulate "human" (source) data through a blue circle pusher and "robot" (target) data through a salmon triangle pusher (Fig. 3), with lower floor friction. The differences aim to analogize the appearance and agent-environment dynamics gaps between human and robot data.

**Source and Target Domain Data.** In our "robot" target domain, we collect demos in the standard push-T setting, but in our "human" source domain, we alter the background color to purple and change the T configuration to be mirrored, requiring a new motion to slot into place 3. The change in background color is analogous to the human data containing new visual scenery, and the change in starting configuration is analogous to the human demonstrating new motions in their demonstrations.

**Training and Evaluation.** To eliminate factor from model design (Sec. 4.3), we choose a standard ResNet-UNet Diffusion Policy [6] and apply the OT-joint loss on the feature outputs of the ResNet encoder. We perform standard action normalization and co-train the policy on both the triangle and

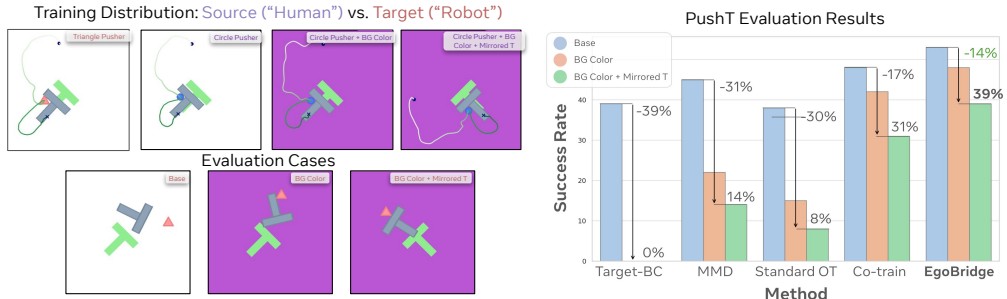

Figure 3: In the simulated Push-T experiments, we probe a toy version of visual and motion level generalization from human to robot. We have narrow target "robot" data represented by the triangle pusher on a white background, and diverse source "human" data represented by the circle pusher with changes in background color and T configuration. We test our "robot" on the diverse human scenarios, and find that EgoBridge outperforms traditional Domain Adaptation baselines.

Table 1: Real World Evaluation Results: In-Distribution and Generalization

| Method | Scoop Coffee (SR) | | | Drawer (SR) | | | Laundry |
|---|---|---|---|---|---|---|---|
| | In-Dist. | Obj. Gen. | Scene+Obj Gen. | Total (Pts \| SR) | Place Toy | Beh. Gen. | (Pts \| SR) |
| Robot-only BC | 33% | 40% | 7% | 38 \| 9% | 28% | 0% | 38 \| 28% |
| Co-train | 53% | 46% | 0% | 55 \| 22% | 42% | 0% | 41 \| 33% |
| EgoMimic [1] | 60% | 53% | 0% | 49 \| 14% | 39% | 0% | 38 \| 33% |
| MimicPlay [25] | 33% | 27% | 0% | 33 \| 14% | 22% | 0% | 32 \| 28% |
| ATM [19] | 47% | 33% | 0% | 56 \| 6% | 17% | 8% | 35 \| 28% |
| **EgoBridge** | **67%** | **60%** | **27%** | **77 \| 47%** | **72%** | **33%** | **48 \| 72%** |

circle pusher data. We evaluate the policy on 3 cases: Triangle in the standard setting, Triangle with purple background, Triangle with purple background and Triangle with purple background and flipped T. We evaluate a total of 100 fixed seeds across all the models and report the mean reward (max IoU with goal) and the success rate (reward $\geq 0.9$).

**Baselines.** In the more controlled simulation settings, we choose to compare EgoBridge against conventional domain adaptation baselines. We choose **Maximum Mean Discrepancy (MMD)** [30] as an alternative domain adaptation loss to joint-OT on the feature encoder. We also test **Standard OT**, which performs marginal alignment instead of joint alignment. The **Co-train** baseline trains on evenly-sampled data from both domains without an alignment loss. Finally, **Target-only** is a control study which trains the policy only on the target (triangle) data.

## 5.2 Real World Evaluation

We evaluate EgoBridge on three challenging real-world manipulation tasks, as illustrated in Fig. 4.

*Drawer:* The robot interacts with a 6x4 drawer array, tasked to pick a toy, place it into a pre-opened drawer, and close it. Robot data (144 demonstrations) covers three of the four array quadrants, each quadrant being a 3x2 arrangement. Human data (1 hour) covers all four quadrants, providing demonstrations for motions into the fourth, robot-unseen quadrant. This setup specifically tests *behavior generalization* to drawer locations only seen in human data. Points (**Pts**) are awarded for successful completion of each stage and a trial is considered a success only if the robot completes all actions. Evaluation uses 48 trials (2 rollouts for each of the 24 drawers).

*Scoop Coffee*: The robot uses its left arm to scoop coffee beans with a spoon and empty them into a target. Robot data (50 demonstrations) involves a specific target (can) in one scene. Human data (2 hours) includes demonstrations with both the can and a new target (grinder), across two distinct scenes, one of which is novel to the robot. Target object positions are randomized (30x23 cm area). We evaluated *observation generalization* for: (1) the new grinder target, and (2) the new scene with the new target, that is, scooping to the grinder in the new scene, seen in human data only. Performance is measured by success rate over 15 rollouts across 5 distinct target locations.

*Laundry:* This is a bimanual task where the robot needs to fold the shirt in 50 x 22 cm range with a rotation range ± 30 degrees. The robot uses both arms to fold the right sleeve, the left sleeve, and

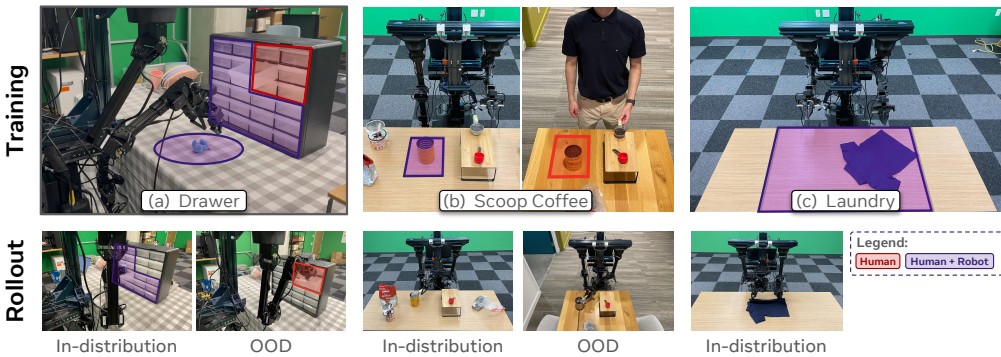

Figure 4: **Training Data and Evaluation Settings.** We show the distribution of human and robot training data (top) and evaluation setting, where in-distribution scenarios are in both human and robot data, while out-of-distribution (OOD) scenarios is seen only in human data.

then the final stage to fold the shirt in half. We award **Pts** for each successful stage and consider it success if all the individual stages are successful. We collect 2 hours of robot data which include 300 demonstrations across 3 shirts, and 2 hours of human data comprising approximately 700 demonstrations. We conduct 18 evaluations with diverse shirt initial placement and colors.

**Baselines.** We adopt the following baselines for real-world tasks. *Co-train:* Direct co-training of the robot and human data using BC loss, without any latent alignment. *EgoMimic: [1]* Co-training with explicit vision and action-space alignment using masking, shared end-effector pose head (human and robot) and a separate joint-space head for robot. *Mimicplay: [25]* A hierarchial policy with a latent high-level planner co-trained on human and robot data, and an action decoder fine-tuned on robot data. *Any-Point Trajectory Modeling (ATM): [19]* A hierarchical policy where the high-level planner is initially co-trained on 2D point tracks derived from both robot and human video data. These point tracksare obtained via Co-tracker [35]. Following this, high-level planner is frozen and an action decoder is fine-tuned specifically on robot data. *Target-only BC:* Trained only on robot data.

## 5.3   Results

**EgoBridge improves in-domain task performance (H1).** Over our set of in-domain tasks, we observed an improvement of 7-44% in absolute task success rate over both human-augmented imitation learning baselines and robot-only behaviour cloning policies. While methods like naive co-training, EgoMimic and ATM also improve in-domain performance across all tasks, EgoBridge consistently outperforms them. We hypothesize that the strongly aligned latent representations facilitate better cross-embodiment transfer.

**EgoBridge enables generalization to objects and scenes only seen in human data (H2).** While it is difficult to collect robot data across diverse scenes and objects, it is trivial to do for human data, so it's critical that we can transfer this knowledge from human to robot, inspiring our experimental setup. Specifically, in the *Scoop Coffee* task, our human data introduces a new coffee grinder, table, lighting and height variations, completely unseen to the robot. We find EgoBridge outperforms all baselines when tested on the new coffee grinder (7-33%). Further, most methods fail entirely when tested on the new grinder + scene, but EgoBridge retains a performance of 27%. We observe similar robustness trends in our simulated benchmark Fig. 3, where EgoBridge enables generalization in the push-T task to a new background and starting configuration, outperforming all baselines.

**EgoBridge enables generalization to new behaviors only seen in human data (H2).** In our most challenging setting, we seek to show that we can learn *entirely new* motions from human data alone. In the drawer task, the robot data covers 3/4 drawer quadrants, whereas the human data covers all 4 quadrants. We evaluate our policy's performance on these new drawers, and find that EgoBridge is able to generalize to these locations with a success rate of 33%, whereas most methods fail entirely (Tab. 1). While all the methods were exposed to the same human data, only EgoBridge was able to effectively transfer the human motion to a novel robot action. We attribute this success to the well aligned latent representations, which enables human to robot knowledge interpolation. We also

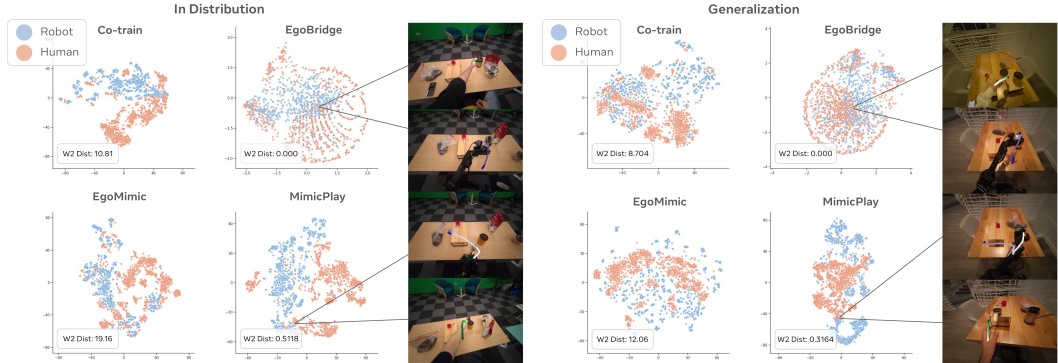

Figure 5: Visualization of TSNE plots on encoded features for EgoBridge and baselines, with the mean Wasserstein-2 distance and KNN pairs of aligned human-robot data visualized.

observed a similar trend in the simulated benchmark where EgoBridge faces the lowest performance drop of 14% compared to all baselines in the *mirrored-T + background colour* reflected in Fig. 3.

**EgoBridge learns a shared latent space that aligns human and robot data in a task relevant manner (H3).** We hypothesize that an ideal latent space for transfer would jointly embed human and robot data into a space with high overlap and semantic interoperability. To probe this, we create a TSNE visualization of the action tokens from our transformer backbone. EgoBridge not only exhibits the highest latent overlap between human and robot as measured by Wasserstein distance as seen in Fig. 5, but also upon inspecting K-nearest neighbor pairs in latent space, exhibits the most semantically similar neighbors. For instance, we see the human and robot performing the same phase of a given task, whereas in baselines like MimicPlay that aligns marginals with KL-div, the semantic similarity is lacking. This result is highly correlated with the task success rates for in-distribution and generalization where baselines with poor alignment perform lower consistently across all evaluations.

**Ablation.** We ablate three key components of our method 1) replacing our DTW-based pairing metric to instead use simple MSE, 2) replacing the joint OT objective $\mathcal{L}_{\text{OT-joint}}$ with standard marginal alignment, and 3) removing any auxiliary alignment objectives (direct co-training). We find that replacing the cost function with MSE leads to the largest performance drop for in-distribution policy success rate from 47% to 17%, seen in Tab. 2, which emphasizes the importance of creating semantically similar pseudo-pairs. Ablating Joint-OT

Table 2: Ablation Results (Drawer)

| Method | Drawer (SR) | Beh. Gen. (SR) |
|---|---|---|
| **EgoBridge** | **47%** | **33%** |
| MSE | 14% | 17% |
| Standard-OT | 33% | 17% |
| Co-train | 22% | 0% |

also shows a large performance drop in both in-distribution and generalization cases which emphasizes how naive marginal alignment cannot transfer knowledge from human data effectively. Ablating an auxiliary alignment loss also shows a significant performance drop for in-distribution success rate and leads to failure in generalization cases, emphasizing the need for joint distribution alignment.

## 6    Conclusion and Limitations

We presented EgoBridge, a novel co-training framework designed to enable robots to learn effectively from egocentric human data by explicitly addressing domain gaps. By leveraging Optimal Transport on joint policy latent feature-action distributions, guided by Dynamic Time Warping cost on action trajectories, EgoBridge successfully aligns human and robot representations while preserving critical action-relevant information. Our experiments demonstrated significant improvements in real-world task success rates (up to 44% absolute gain) and, importantly, showed robust generalization to novel objects, scenes, and even tasks observed only in human demonstrations, where baselines often failed.

While EgoBridge demonstrates promising single-task transfer, the DTW-based action alignment cost may not be as informative in multi-task joint domain adaptation settings. In future works, we seek to adopt more generalizable alignment cost such as natural language embedding distances [36] obtained from Vision-Language models and visual features extracted from foundation models. Other avenues for future work include extending the joint distribution adaptation to multiple embodiments and leveraging Internet-sourced human data without action labels.

# 7 Acknowledgment

This work was supported in part by a research gift from Meta Platforms, Inc, the Defense Advanced Research Projects Agency (DARPA) under the TIAMAT program and by the National Science Foundation under Grant No. 2144194. Any opinions, findings, and conclusions or recommendations expressed in this material are those of the author(s) and do not necessarily reflect the views of the supporting organizations.

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

# A  Algorithm Pseudocode

**Overview.** Algorithm 1 describes the joint policy co-training of the human and robot data with the joint OT loss.

**Mini-batch sampling.** At each iteration we draw two size–$B$ mini-batches, one from the human dataset $\mathcal{D}_H$ and one from the robot dataset $\mathcal{D}_R$ (Lines 3–4).

**Shared encoder.** Both batches are fed through a *shared* encoder $f_\phi$ that maps raw observations $o$ to a latent embedding $z \in \mathbb{R}^d$ (Line 6).

**Algorithm 1** EgoBridge Co-Training for Human $\leftrightarrow$ Robot Imitation

---

**Require:** Human demos $\mathcal{D}_H$, robot demos $\mathcal{D}_R$, OT loss weight $\alpha$, DTW discount $\lambda$, Sinkhorn regularization $\varepsilon$, LR schedule $\eta$
**Ensure:** Encoder $f_\phi$ and policy decoder $\pi_\theta$
 1: Initialize network parameters $\phi, \theta$
 2: **while** training not converged **do**
 3:      $\mathcal{B}_H = \{o_H^{(i)}, a_H^{(i)}\}_{i=0}^B \leftarrow$ SAMPLEBATCH$(\mathcal{D}_H, B)$               ▷ sample human data
 4:      $\mathcal{B}_R = \{o_R^{(i)}, a_R^{(i)}\}_{i=0}^B \leftarrow$ SAMPLEBATCH$(\mathcal{D}_R, B)$               ▷ sample robot data
 5:      $z_H \leftarrow f_\phi(o_H), z_R \leftarrow f_\phi(o_R)$      ▷ encode raw observation batch into shared latent space
 6:      **for all** $i \in \{0, \ldots, |B_H|-1\}$, $j \in \{0, \ldots, |B_R|-1\}$ **do**
 7:          $A_{ij} \leftarrow$ DTW$(a_H^{(i)}, a_R^{(j)})$                ▷ DTW distance between actions
 8:      **end for**
 9:      $i^*(j) \leftarrow \arg\min_i A_{ij} \,\forall j \in \{0, \ldots, |B_R|-1\}$    ▷ pick the most similar human traj. for each robot traj.
10:      **for all** $i \in \{0, \ldots, |B_H|-1\}$, $j \in \{0, \ldots, |B_R|-1\}$ **do**
11:          $D_{ij} \leftarrow \|z_H^{(i)} - z_R^{(j)}\|_2^2$                 ▷ observation feature cost
12:          **if** $i = i^*(j)$ **then**
13:              $\tilde{C}_{ij} \leftarrow \lambda D_{ij}$             ▷ discount cost if DTW says the pair matches
14:          **else**
15:              $\tilde{C}_{ij} \leftarrow D_{ij}$
16:          **end if**
17:      **end for**
18:      $\mu_H \leftarrow \frac{1}{|\mathcal{B}_H|}\mathbf{1}$, $\mu_R \leftarrow \frac{1}{|\mathcal{B}_R|}\mathbf{1}$      ▷ uniform probability mass on the minibatch supports
19:      $T^* \leftarrow$ SINKHORN$(\mu_H, \mu_R, \tilde{C}, \varepsilon)$      ▷ compute differentiable OT loss with entropic reg.
20:      $\mathcal{L}_{\text{OT-joint}} \leftarrow \sum_{i,j} T_{ij}^* \tilde{C}_{ij}$
21:      $\hat{a}_H \leftarrow \pi_\theta(z_H)$, $\hat{a}_R \leftarrow \pi_\theta(z_R)$             ▷ predict actions for both domains
22:      $\mathcal{L}_{\text{BC-cotrain}} \leftarrow \mathcal{L}_{\text{BC}}(\hat{a}_H, a_H) + \mathcal{L}_{\text{BC}}(\hat{a}_R, a_R)$      ▷ behaviour-cloning loss across domains
23:      $\mathcal{L} \leftarrow \mathcal{L}_{\text{BC-cotrain}} + \alpha\,\mathcal{L}_{\text{OT-joint}}$
24:      $(\phi, \theta) \leftarrow$ OPTIMIZERSTEP$(\phi, \theta, \nabla_{\phi,\theta}\mathcal{L}, \eta)$
25: **end while**
26: **return** $f_\phi, \pi_\theta$

---

**Behavioural pairing via DTW.** The Dynamic Time Warping (DTW) cost is applied on the *action* sequences $a_H^{(i)}$ and $a_R^{(j)}$ from both domains. The assignment $i^*(j)$ (Line 11) identifies a single pseudo-pair in $\mathcal{D}_H$ for each robot trajectory $j$ which minimizes the DTW cost.

**Sinkhorn.** The Sinkhorn-Knopps algorithm is used to compute an entropy-regularized Optimal Transport plan from the uniform probability mass on the minibatch supports.

**Joint-OT.** Latent distances $D_{ij} = \|z_H^{(i)} - z_R^{(j)}\|_2^2$ are discounted by $\lambda < 1$ when $(i, j)$ is behaviourally matched (Lines 12–17), producing a shaped cost $\tilde{C}$ that biases the OT loss computed from the Sinkhorn OT transport plan.

**Behaviour cloning and objective.** The decoder $\pi_\theta$ produces domain-specific actions and is trained with a supervised loss $\mathcal{L}_{\text{BC-cotrain}}$ (Lines 22–24). The overall objective (Line 26)

$$\mathcal{L} = \mathcal{L}_{\text{BC-cotrain}} + \alpha\,\mathcal{L}_{\text{OT-joint}}$$

balances imitation fidelity with latent-space alignment, with an $\alpha$ balancing factor between the two losses.

## B   Network Architecture

The following section describes how $f_\phi$ and $\pi_\theta$ are parameterized in the simulation and real-world experiments. All hyperparameters are summarized in Table 3, Table 4 for real-world and in Table 5 for simulation.

## B.1 Real World Experiments

Table 3: Hyperparameters for Real-World Experiments

| Symbol | Value |
|---|---|
| $H \times W \times C$ | $480 \times 640 \times 3$ |
| $d_{\text{proj}}$ | 256 |
| $D_{\text{stem}}$ | 8 |
| $d_{\text{attn}}$ | 64 |
| $L$ | 16 |
| $d_q(Robot)$ | 7 or 14 (joint positions) |
| $d_q(Human)$ | 6 or 12 (end effector pose as xyz + euler) |
| $D_{\text{trunk}}$ | 8 |
| $N_{\text{trunk}}$ | 16 |
| $d$ | 256 |
| $M$ | 8 |
| $D_{\text{head}}$ | 8 |
| $N_{\text{head}}$ | 8 |
| $d_{\text{head}}$ | 64 |
| $k$ | 100 |
| $d_a$ | 7 or 14 (xyz + euler angles + gripper position) |
| $L_{BC}$ (Robot) | $SmoothL1(\text{xyz}) + SmoothL1(\text{gripper}) + 0.5 \cdot MSE(\text{euler})$ |
| $\mathcal{L}_{\mathcal{BC}}$ (Human) | $SmoothL1(\text{xyz})$ |
| $\mathcal{L} = \mathcal{L}_{\texttt{BC-cotrain}} + \alpha\, \mathcal{L}_{\texttt{OT-joint}}$ | $\alpha = 0.7$ |

### B.1.1 Observation Encoder: $f_\phi$

$f_\phi$ consists of two components, "stems" for each observation modality and a "trunk". The stems are shallow networks that encode heterogenous input spaces into a fixed representation to allow joint learning between human and robot observations.

**Vision Stem.** Given an RGB frame $\mathbf{I} \in \mathbb{R}^{H \times W \times 3}$, we normalize using ImageNet normalization and then pass it through a ResNet-18 encoder truncated *before* the global-pool layer to obtain a $7 \times 7 \times 512$ feature map. The 49 spatial patches are flattened and projected with a single-layer MLP of hidden dimension $d_{\text{proj}}$, producing 49 tokens of size $d_{\text{proj}}$ per image. A single multi-head cross-attention block ($D_{stem}$ heads, per-head width $d_{\text{attn}}$) employs $L$ learnable query tokens of dimension $d$ that attend to these 49 patch tokens; the resulting $L$ query outputs constitute the vision tokens passed to the shared trunk.

**Proprioceptive Stem.** The proprioceptive observation vector $\mathbf{q} \in \mathbb{R}^{d_q}$ (joint angles, end effector pose etc.) is z-score normalized and passed through a single-layer MLP of hidden dimension $d_{\text{proj}}$, producing one token $k_{\text{proprio}} \in \mathbb{R}^{d_{proj}}$. A single multi-head cross-attention block ($D_{stem}$ heads, per-head width $d_{\text{attn}}$) uses the $L$ learnable query tokens as in the vision branch to attend to this key/value token; the resulting $L$ query outputs constitute the proprioceptive tokens that are forwarded to the shared trunk.

**Trunk.** The *shared* "trunk" is a standard multi-block transformer encoder, which consists of $D_{trunk}$ heads, $N_{trunk}$ blocks, and an embedding dimension of $d$. Each head has an attention dimension of $d/D_{trunk}$. A token sequence of length $L \cdot m$ by concatenating the token outputs of $m$ observation encoding stems. A set of $M$ learnable context tokens are prepended to the the token sequence. The new sequence of $M + m \cdot L$ are input into the trunk. The first $M$ are extracted from the output sequence and represent the feature output $z$ of $f_\phi$, and consequently where $\mathcal{L}_{\textbf{OT-joint}}$ is applied.

### B.1.2 Policy: $\pi_\theta$

$\pi_\theta$ is conditioned on the output $z$ of $f_\phi$ and is parameterized by a DETR-style multi-block transformer decoder "head" [34].

**Head.** The head consists of $N_{head}$ blocks and $D_{head}$ heads, a cross-attention dimension of $d_{head} \cdot D_{head}$ and a self-attention dimension of $d_{head}/D_{head}$. This hybrid attention is designed to promote increased conditioning from the latent $z$. $k$ learnable tokens of $d_{head}$ are input into the

model, where $K$ corresponds to the action chunk length $k$. Each block in the decoder consists of alternating self-attention on the $k$ input tokens, and cross-attention with the context $M$ tokens ($z$) from the trunk. After the final block, a linear layer projects the output to $d_a$ corresponding to the cartesian action dimension, which is described precisely in Section D. Since the human data only comprises of 3D position, the loss is a masked loss which is only calculated on the values in the prediction corresponding to the 3D position for human data. The loss $\mathcal{L}_{\mathcal{BC}}$ on human data is Smooth L1 loss on the 3D position, while the loss $\mathcal{L}_{\mathcal{BC}}$ on robot data is Smooth L1 loss on the 3D position, gripper value and MSE loss on the orientation.

**Complete Architecture.** The network consists of a *shared* vision stem processes main egocentric RGB images ($I_{ego} \in \mathbb{R}^{H \times W \times 3}$) from both human and robot and produces $L$ tokens. Followed by this are individual *embodiment-specific* proprioceptive stems that map $q^H$ and $q^R$ proprioceptive features to an additional $L$ tokens. To the vision and proprioceptive tokens, left and right wrist RGB images ($I_{wrist} \in \mathbb{R}^{H \times W \times 3}$) are mapped to additional tokens by one additional wrist vision stem for single-arm robot data and two additional wrist vision stems for bimanual robot data. $M$ learnable tokens are prepended to the token sequence and passed to the trunk. The OT-joint loss is applied on the first $M$ tokens of the output. The same tokens are decoded by the head into $a_{t:t+k} \in SE(3) + gripper$, where an action for a single arm $\in R^{k \times 7}$ with the orientation expressed as euler angles in the yaw-pitch-roll convention. For bimanual tasks, the action comprises of two such action trajectories concatenated resulting in an output $\in R^{k \times 14}$.

**Training Details.** We train the real world EgoBridge model on a single L40s gpu for 100000 iterations on the Drawer task, 110000 iterations on the Laundry task, and 120000 iterations on the Scoop Coffee task, which takes about 24 hours. More details are in Table 4.

Table 4: Training Details for Real World Experiments

| Parameter | Value |
|---|---|
| Optimizer | AdamW |
| Learning Rate | $5 \times 10^{-5}$ |
| Weight Decay | 0.0001 |
| Scheduler | Linear |
| Batch Size | 32 |
| Data Augmentations | ColorJitter + ImageNet Normalization (ResNet) |
| OT Loss | GeomLoss [37] |
| Blur | 0.05 |
| Distance | Sinkhorn |

### B.2 Sim PushT

**Data Processing.** We follow the PushT environment setup introduced in the Diffusion Policy benchmark suite [6]. Each demonstration comprises a sequence of RGB observations and associated low-dimensional proprioception, along with corresponding action labels. The RGB observations $I_t \in \mathbb{R}^{96 \times 96 \times 3}$ are normalized using ImageNet statistics. The proprioception consists of 2D end-effector positions $(x, y)$, which are min-max normalized per-dimension using statistics computed over the entire dataset. The action at each timestep is defined as the target 2D position of the end-effector, also normalized with per-dimension min-max normalization. Action chunks of length $k = 16$ are extracted from each demonstration trajectory. No observation history is used in the input, i.e., the observation context window is 1.

**Encoder ($f_\phi$).** We employ the same vision encoder architecture as in the original Diffusion Policy paper [6], which uses a truncated ResNet-18 to extract spatial feature maps from $96 \times 96 \times 3$ RGB images. The proprioceptive input (normalized $x, y$ position) is concatenated channel-wise to the image feature maps before being passed as the global conditioning to the UNet-based diffusion decoder. This architecture has been shown to effectively fuse visual and proprioceptive features for policy learning in low-dimensional manipulation tasks like PushT. The Joint-OT loss is applied on the image-proprio concatenated feature output.

**Policy ($\pi_\theta$).** The policy architecture mirrors the conditional diffusion policy design [6]. The output is a denoising network trained to generate a trajectory segment of length $k = 16$ for 2D end-

effector coordinates in normalized space. During training, the policy receives a noisy version of the ground-truth action chunk, and learns to iteratively denoise the sample using the concatenated latent observations. The diffusion loss is the weighted MSE over all denoising steps in the sampling process. During inference, the policy performs iterative denoising starting from Gaussian noise to generate 2D trajectory samples conditioned on the current observation.

**Training Details.** We train the simulation EgoBridge model on a single A40 GPU for 130000 iterations, which corresponds to around 2 hours of training time. The hyperparameters and training details are summarized in Table 5.

Table 5: Training Details for Simulation Experiments (PushT)

| Parameter | Value |
|---|---|
| Optimizer | AdamW |
| Learning Rate | $1 \times 10^{-4}$ |
| Weight Decay | $1 \times 10^{-6}$ |
| Scheduler | Cosine |
| Warmup Steps | 500 |
| Iterations | 130,000 |
| Batch Size | 32 |
| Exponential Moving Average (EMA) | Power = 0.75 |
| Data Augmentations | ImageNet Normalization (ResNet) |
| $\mathcal{L}_{\texttt{BC}}$ (Triangle & Circle) | $MSE(\text{xy})$ |
| $\mathcal{L} = \mathcal{L}_{\texttt{BC-cotrain}} + \alpha\,\mathcal{L}_{\texttt{OT-joint}}$ | $\alpha = 0.2$ |
| OT Loss | GeomLoss [37] |
| Blur | 0.01 |
| Distance | Sinkhorn |

# C  Baselines

## C.1  Real World Experiments

All baselines instantiate the encoder $f_\phi$ and decoder $\pi_\theta$ architecture described in Section B and Table 3 with modifications to input and output spaces.

**Robot-only Behavioral Cloning (Robot-only BC)**. This baseline is trained only on robot observations. The encoder includes vision stems for egocentric and wrist camera inputs and a proprioceptive stem for joint-space inputs. The resulting tokens and $M$ context tokens are passed to the transformer trunk. The decoder head uses $K$ learnable tokens to produce future actions, where each action is a 7-dimensional Cartesian command: 3D position, 3D orientation, and scalar gripper. The loss is computed using Smooth L1 for position and gripper, and MSE for orientation.

**Co-train**. Co-train uses both human and robot demonstrations during training and represents an ablated version of the EgoBridge model, where the joint-OT loss ($\mathcal{L}_{\text{OT-joint}}$) is not applied.

**EgoMimic [1]**. EgoMimic follows the Co-train architecture but includes two modifications: (1) The egocentric RGB input is masked with an overlay and red line augmentation, following [1] before it is input into the encoder, and (2) The decoder uses two heads: one for robot joint-space commands $\in R^7$ and another for shared cartesian pose $\in SE(3) + gripper$ for human and robot data. The joint-space predictions are supervised using a Smooth L1 loss on the entire prediction vector, while cartesian pose predictions are supervised by the loss described in Section B. During training, the appropriate head is selected based on the data source, where robot data provides gradient updates through the joint predictions and cartesian predictions, while the human data provides gradient updates for the model through the cartesian predictions. The identical overlay is applied during evaluation.

**ATM [19].** For fair comparisons, we implement ATM with an identical architecture as our method and extend it with an additional head designed to predict future dense 2D motion trajectories in the image plane, as described in the original ATM paper. The goal is to test whether access to pixel-level dynamics improves downstream behavior cloning performance via improved latent structure.

Ground-truth 2D pixel tracks are generated using CoTracker [35] and organized into a temporal stack of length 10, covering 100 future frames. The action-track head outputs future 36 keypoints across $6 \times 6$ grid over the image plane. Training proceeds in two stages: first, the the trunk, stems, and the 2D track head are pre-trained for 500 epochs to jointly predict human and robot 2D action tracks; second, the trunk and 2D head are frozen, and only the cartesian action head is fine-tuned on robot data. We opt for stacked, non-autoregressive, predictions of the full temporal keypoint trajectory, diverging from the sequential autoregressive formulation in the original ATM [19], in order to isolate the representational benefits of point-track conditioning on downstream imitation performance.

**MimicPlay [25].** MimicPlay is implemented as a hierarchical policy which consists of a "high-level" policy which is co-trained on both human and robot data and a "low-level" policy which is trained only on robot data and is conditioned on the context output of the "high-level" policy.

*High-level policy.* We parameterize the high level policy of MimicPlay with a *shared* vision stem for $I_{ego}$ for both human and robot and a shallow "trunk" which consists where $N_{trunk} = 2$, while the other hyperparameters for the trunk remain the same to match the parameter size of the original high-level in MimicPlay. 2 tokens corresponding to the high level context are prepended to the vision tokens. The first 2 output tokens are the latent context $z_{high}$ that are passed to the low-level policy. A KL-divergence loss is applied between the robot data $z_{high}^{R}$ and the human data $z_{high}^{H}$. A Gaussian Mixture Model decoder with identical parameters to the MimicPlay paper, takes $z_h igh$ and predicts the mixing coefficient, mean, and the variance of GMM. With these predictions, a sequence of 100 3D positions are sampled from the GMM for a given input observation and supervised with the ground-truth actions for the observation with a negative log likelihood loss. The high-level planner is trained until convergence which is around 60000 iterations.

*Low-level policy.* The low level policy is parameterized the Robot-only BC model. The high-level policy is frozen and the latent output $z_{high}$ is concatenated to the $M$ tokens produced by $f_\phi$ which are then decoded into cartesian actions supervised by $\mathcal{L}_{BC}$ on robot data.

## C.2 Sim PushT

In the more controlled Sim PushT environment, we benchmark EgoBridge against standard domain adaptation techniques commonly used in feature-level alignment. All baselines instantiate the same encoder $f_\phi$ and decoder $\pi_\theta$ described in Section B.2, with differences only in the training objective and data source used.

**Maximum Mean Discrepancy [30].** We evaluate domain alignment using the MMD loss [30] as an alternative to the Joint OT loss. This baseline uses the same training configuration as EgoBridge but replaces the Joint-OT objective with the MMD loss applied on the latent embeddings from the shared encoder. The overall training objective is:

$$\mathcal{L} = \mathcal{L}_{\text{BC-cotrain}} + \lambda_{\text{MMD}} \cdot \mathcal{L}_{\text{MMD}}, \quad \text{where} \quad \lambda_{\text{MMD}} = 1.0$$

The MMD loss is computed as:

$$\mathcal{L}_{\text{MMD}} = \frac{1}{n^2} \sum_{i,j} k(z_H^{(i)}, z_H^{(j)}) + \frac{1}{m^2} \sum_{i,j} k(z_R^{(i)}, z_R^{(j)}) - \frac{2}{nm} \sum_{i,j} k(z_H^{(i)}, z_R^{(j)})$$

where $k(\cdot, \cdot)$ is a Gaussian RBF kernel:

$$k(x, y) = \exp\left(-\frac{\|x - y\|_2^2}{2\sigma^2}\right)$$

and $\sigma$ is the kernel bandwidth hyperparameter.

**Standard OT.** This baseline implements marginal alignment using an unshaped Optimal Transport cost matrix without any DTW-based behavioural pairing. It follows the same formulation as Ego-Bridge (see Algorithm 1) but removes the DTW pairing step (Lines 11–17), using direct squared $\ell_2$ distance $\tilde{C}ij = Dij$ for all $(i, j)$ in the batch. This serves as an ablation that tests the benefit of behaviourally guided pairings in EgoBridge.

**Co-train.** Similar to the real-world setting, the co-train baselines ablates the Joint OT loss from the sim EgoBridge architecture to evaluate the baseline performance of training a shared policy on both embodiments.

**Target-only BC.** This baseline is trained using only robot demonstrations from the triangle pusher in the white background with original T configuration (100 trajectories total). It uses the same architecture as EgoBridge but is never exposed to data from the circle pusher (source domain).

## D  Robot Data

**Bimanual Manipulator.** To effectively utilize egocentric human data for manipulation, the robot hardware platform must resemble human sizes and kinematic workspaces. Drawing inspiration from the "Eve" robot platform introduced in EgoMimic [1], we develop a custom mobile manipulator that comprises of two 6-DoF ViperX 300s mounted in an identical inverted configuration on a height-adjustable rig. Similar to Eve, we propose to leverage the Project Aria glasses [32] as the main egocentric perception sensor for the robot and mount it in a way that emulates the hand-eye configuration of a human adult. This effectively mitigates the human-robot camera device gap and reduces the sensor-manipulator kinematic gap. Each arm is equipped with an Intel Realsense D405 on its wrist to facilitate precise near-range manipulation. The raw RGB data from the aria glasses is undistorted to a linear camera model, resized and reshaped to $480 \times 640 \times 3$. Fig. 6 shows an annotated picture of the bimanual manipulator. The bimanual manipulator is teleoperated using a leader-follower system similar to ALOHA [5], where two WidowX 6-DoF arms in an inverted configuration as seen in Fig. 6

**Data Processing.** Each robot demonstration comprises synchronized streams of joint-space proprioception, RGB image observations, and action labels recorded at 50Hz. At each timestep $t$, we collect the current joint configuration $\mathbf{q}_t^R \in \mathbb{R}^{d_q}$, where $d_q$ denotes the number of actuated joints for one or two robot arms. RGB images are recorded from two sources: (1) an egocentric camera ($I_{ego,t}^R \in \mathbb{R}^{480 \times 640 \times 3}$) mounted using Project Aria glasses in a head-like configuration, and (2) wrist-mounted RGB cameras ($I_{wrist,t}^R \in \mathbb{R}^{480 \times 640 \times 3}$) placed near the end-effectors.

The raw action at each timestep is defined as the commanded joint position $\mathbf{a_r aw}_t^R \in \mathbb{R}^{d_q}$, representing the control input issued at time $t$. To obtain cartesian actions used for training, we use analytical forward kinematics to compute the $SE(3)$ cartesian pose. The cartesian pose is then projected into the image frame via the Aria glasses extrinsics ($T_R^{aria}$) obtained through hand-eye calibration, which represent the transformation between the robot base frame and the Aria glasses camera frame. The ground-truth actions $\mathbf{a}_t^R \in \mathbb{R}^{d_q}$ are obtained through

$$\mathbf{a}_t^R = \left(T_R^{\text{aria}}\right)^{-1} \cdot \text{FK}(\mathbf{a}_{\text{raw},t}^R)$$

## E  Embodied human data

We use the Project Aria [32] glasses to collect *human embodied experience data*. The glasses are accompanied by the Aria Machine Perception Services (MPS). The raw data from the Aria glasses consist of an RGB camera stream, SLAM cameras, IMU, eye cameras. This raw data is uploaded to a cloud service provided by Project Aria, known as the MPS. The service returns device pose estimated using the SLAM camera, hand tracking relative to the device frame, a semi-dense point-cloud of the environment and eye gaze estimation. This processed data is obtained as CSV files with aligned timestamps. The hand tracking is in the form of 3D positions $p^H \in SE(3) \times SE(3)$ for both hands in device frame, while the head pose $T^{\text{aria}} \in SE(3)$ in world frame. Each hand position $p_t^H$ is in device frame $T_t^{aria}$ for timestep $t$. The raw RGB data is undistorted to a linear camera model and reshaped to $480 \times 640 \times 3$. **Constructing human actions.** One challenge with unifying the reference frames for joint policy learning is that robot actions are typically in a fixed reference frame (usually the base frame), while human hand tracking is in the device frame, which moves. Following the idea of predicting action chunks [6, 5], we aim to construct action chunks $a_{t:t+k}^H$. Taking the single-arm case without loss of generality, the raw trajectory is $SE(3)$ poses $[p_t^H, p_{t+1}^H, p_{t+2}^H, ..., p_{t+k}^H]$, where each hand position $p_t^H$ is in device frame $T_t^{aria}$. We choose to create *pseudo-reference frames* taking inspiration from [1], where we construct an action $a_{t:t+k}^H$ by projecting $k$ future hand positions into the device frame at timestep $t$. This allows the policy to predict trajectories with respect to a stable reference at each timestep. As such, the trajectory is constructed by:

$$a_{t:t+k}^H = \left[ \left(T_t^{\text{aria}}\right)^{-1} T_{t+i}^{\text{aria}} \cdot p_{t+i}^H \right]_{i=1}^k$$

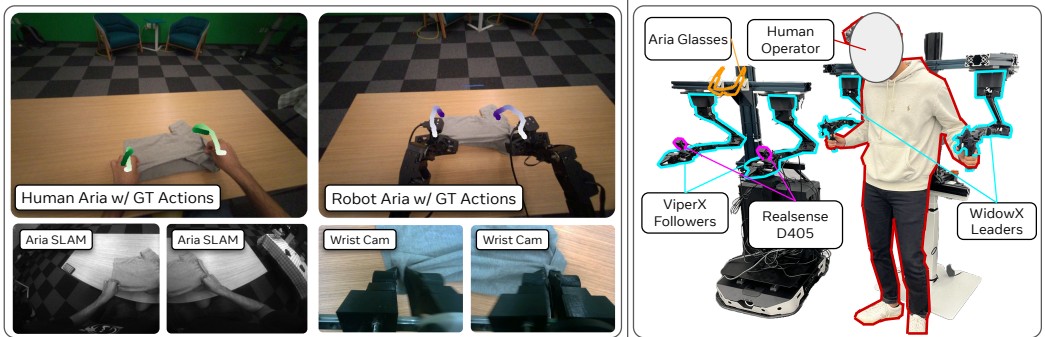

Figure 6: We employ the Aria glasses to capture Egocentric RGB images for both human and robot embodiments. The Aria uses its side SLAM cameras to estimate device pose and hand tracking (left). Ground truth (GT) cartesian actions for both embodiments are projected to pixel space using Aria intrinsics and visualized on the images. Our robot system is a leader-follower system with two Viper X arms as the followers and two Widow X arms as the leaders which are teleoperated by the human demonstrator (right).

## F    Data processing and alignment.

Recent cross-embodiment works  [1, 38, 10] show that individually normalizing proprioception and actions for both embodiments, helps co-training. Similarly, we employ *z-score* normalization by subtracting the per-embodiment dataset mean and dividing by the standard deviation

$$norm(p) = (p - \mu_p)/\sigma_p$$

We normalize actions identically, using per-dimension mean and standard deviation.

For both human and robot, our action chunk size $k = 100$. Human data sensor streams (RGB, hand tracking and SLAM) are received at 30hz. We construct a human action chunk by taking 10 sample pose values at an interval of 3 frames, and interpolating them to a trajectory of length 100. This corresponds to a trajectory of length 100 to 0.9 seconds in real time. We construct a robot action chunk by concatenating the ground-truth actions of 100 successive timesteps which correspond to 2 seconds in real time. The human and robot data sources are visualized in Fig. 6.

## G    Additional Task Details

### G.1    Real World Experiments

**Drawer.**  For this task, we collect one hour of human data and 30 minutes of robot data comprising 144 demonstrations. The human data was equally distributed across all four quadrants, while the robot demonstrations were equally divided among three quadrants (excluding the top-right), with each receiving 48 robot demonstrations (8 demonstrations per drawer). The main failure modes for this task are the inability to correctly grasp the toy, taking the toy towards the incorrect drawer and being unable to completely push the drawer shut.

**Scoop Coffee.**  The scoop coffee task emphasizes observation generalization with novel objects and scenes. To address this, we gathered 180 minutes of human data, equally split across three scenarios: the base scene, a new target object (grinder), and a new target object (grinder) with a new scene. For robot data, we collected 50 demonstrations with the original target object and scene configuration. The two primary failure modes in this task is an inability to precisely grasp the scoop and imprecise targeting of the target container.

**Laundry.**  The laundry task demands precise bimanual coordination. We collected a similar amount of human data (2 hours) as the scoop task, comprising 700 demonstrations, alongside two hours of robot data (300 demonstrations). Common failure modes involve missing one or both sleeves during any of the three sub-stages, leading to an imprecise or unsuccessful fold. The qualitative task successes are visualized in Fig. 7 and common failure modes for each task are visualized in Fig. 8, while the data mixture is shown in Table 6.

a) Drawer

b) Scoop Coffee

c) Laundry

Figure 7: Qualitative success of EgoBridge on each of our real world tasks.

a) Drawer - Failure Modes

b) Scoop Coffee - Failure Modes

c) Laundry - Failure Modes

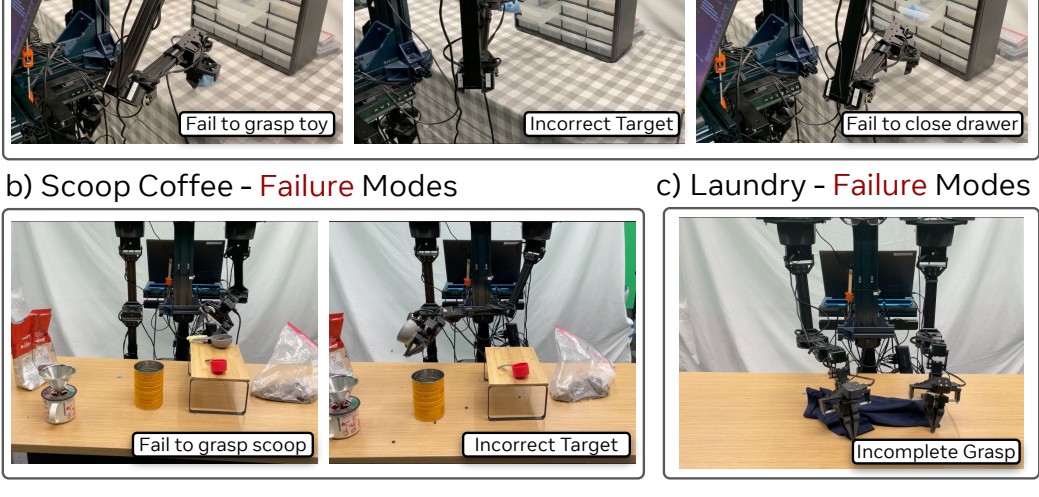

Figure 8: Common failure modes for each of the real world evaluation tasks.

Table 6: Data collection overview for both Human(H) and Robot(R) data. We report both the number(#) of total task demonstrations and the time(min) took to collect them.

| Task | H # | H min | H #/min | R # | R min | R #/min |
|------|-----|-------|---------|-----|-------|---------|
| Drawer | 360 | 60 | 6 | 144 | 29 | 5 |
| Scoop Coffee | 720 | 180 | 4 | 50 | 13.3 | 3.75 |
| Laundry | 700 | 120 | 5.8 | 300 | 120 | 2.5 |

### G.2 Sim PushT

We collect a total of 350 demonstrations across 4 scenarios using the PyMunk simulator environment published with Diffusion Policy [6]. We collect 100 demonstrations each for the circle pusher and triangle pusher in the original T configuration with the white background. In addition, we collect 50 demonstrations each of the circle pusher with original T configuration on purple background, mirrored T on purple background and mirrored T on white background. We apply a friction coefficient multiplier of 0.7 on the triangle pusher to embody a kinematic gap like human and robot.

## H  Supplementary Simulation Results

We evaluate 3 settings : In distribution, which corresponds to the triangle pusher with the original T configuration on a white background; Observation Generalization which corresponds to the previous setting with a purple background; Observation and Behaviour Generalization which corresponds to the triangle pusher with a mirrored T on a purple background. For evaluation, we select a total of 100 seeds between 101 and 9999 using a deterministic random sampling with seed 42. The complete results are summarized in Table 7. The Mean Reward is calculated using the average max Intersection-over-Union with the goal across 100 seeds and Success Rate (SR) is computed using the number of seeds with a reward of over 0.9.

Table 7: Sim PushT results across 3 evaluation settings

| Method | In-distribution (Mean Reward \| SR) | Generalization | |
|--------|-----------------|-----------------|-----------------|
| | | **Purple Bg** (Mean Reward \| SR) | **Purple Bg + Mirrored T** (Mean Reward \| SR) |
| **EgoBridge** | **0.7605 \| 53.00%** | **0.7206 \| 48.00%** | **0.6520 \| 39.00%** |
| Target Only | 0.5555 \| 39.00% | 0.0904 \| 0.00% | 0.0992 \| 0.00% |
| Cotrain | 0.7062 \| 48.00% | 0.6899 \| 42.00% | 0.6214 \| 31.00% |
| Standard OT | 0.7009 \| 38.00% | 0.5303 \| 15.00% | 0.5109 \| 8.00% |
| MMD | 0.6439 \| 45.00% | 0.4867 \| 22.00% | 0.5876 \| 14.00% |

## I  Pseudo-pair Visualisation

To show why Dynamic Time Warping (DTW) is an ideal cost function to identify behaviourally similar trajectories to align, we visualize randomly sampled mini-batch pairs from the dataset for each of the tasks and qualitatively compare them with the Mean Square Error (MSE) cost function pairs used for the MSE ablation. A pair, as defined earlier, is the row-wise cost function minimum in the cost matrix computed from a batch of human and robot data. Qualitatively, DTW is more robust to temporal shifts, viewpoint shifts and is more spatially precise when compared to MSE. Specifically, MSE often pairs the incorrect sub-task in tasks like Laundry, and loses precise spatial location of the target container in tasks like Drawer and Scoop Coffee. The few failure modes of DTW are where the trajectory overlaps a few different stages which leads to temporal misalignments and when locations are spatially apart but visually close due to an extreme viewpoint shift. Despite this, DTW enables pairings that are able to capture very fine-grained changes in trajectory to find the most behaviorally similar between human and robot data. This is visualized in Fig. 9.

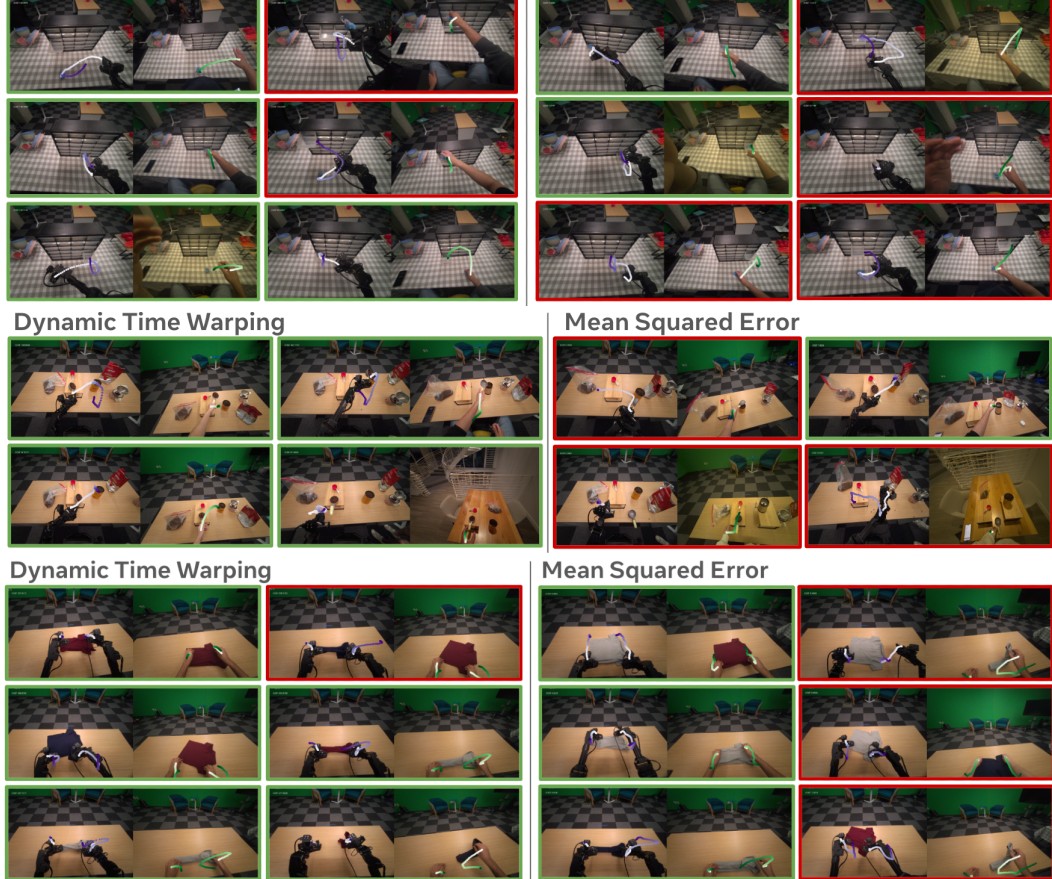

Figure 9: Qualitative visualization of randomly sampled mini-batch pairing of MSE and DTW.

## J  Supplementary Real Results

In order to test EgoBridge's ability to generalize to new appearances under a more challenging real-world setting, we conduct two additional evaluation experiments on the **Laundry** task. The first, *Observation Generalization*, introduces a new white table surface unseen in robot data, along with 30 minutes of human data recorded on this new setup. The second, *Combinatorial Observation Generalization*, evaluates a harder scenario where the robot must handle a new shirt color on the white surface, with that shirt seen only in the human data and excluded entirely from robot demonstrations. Both evaluations follow the same protocol as the main experiments, performing 20 rollouts across 5 randomly sampled workspace positions. As summarized in Table 8, EgoBridge achieves substantially higher success rates and sub-task points than the Co-train baseline in both settings, demonstrating stronger robustness to visual appearance shifts and compositional variations in object–scene configurations.

Table 8: **Generalization experiments on Laundry.** Evaluation of Co-train and EgoBridge under two generalization settings. We report success rate (SR) and sub-task points.

| Model | Obs. Generalization | | Comb. Obs. Generalization | |
| --- | --- | --- | --- | --- |
| | SR (%) | Points | SR (%) | Points |
| Co-train | 25 | 25 | 5 | 7 |
| **EgoBridge** | **50** | **43** | **30** | **35** |

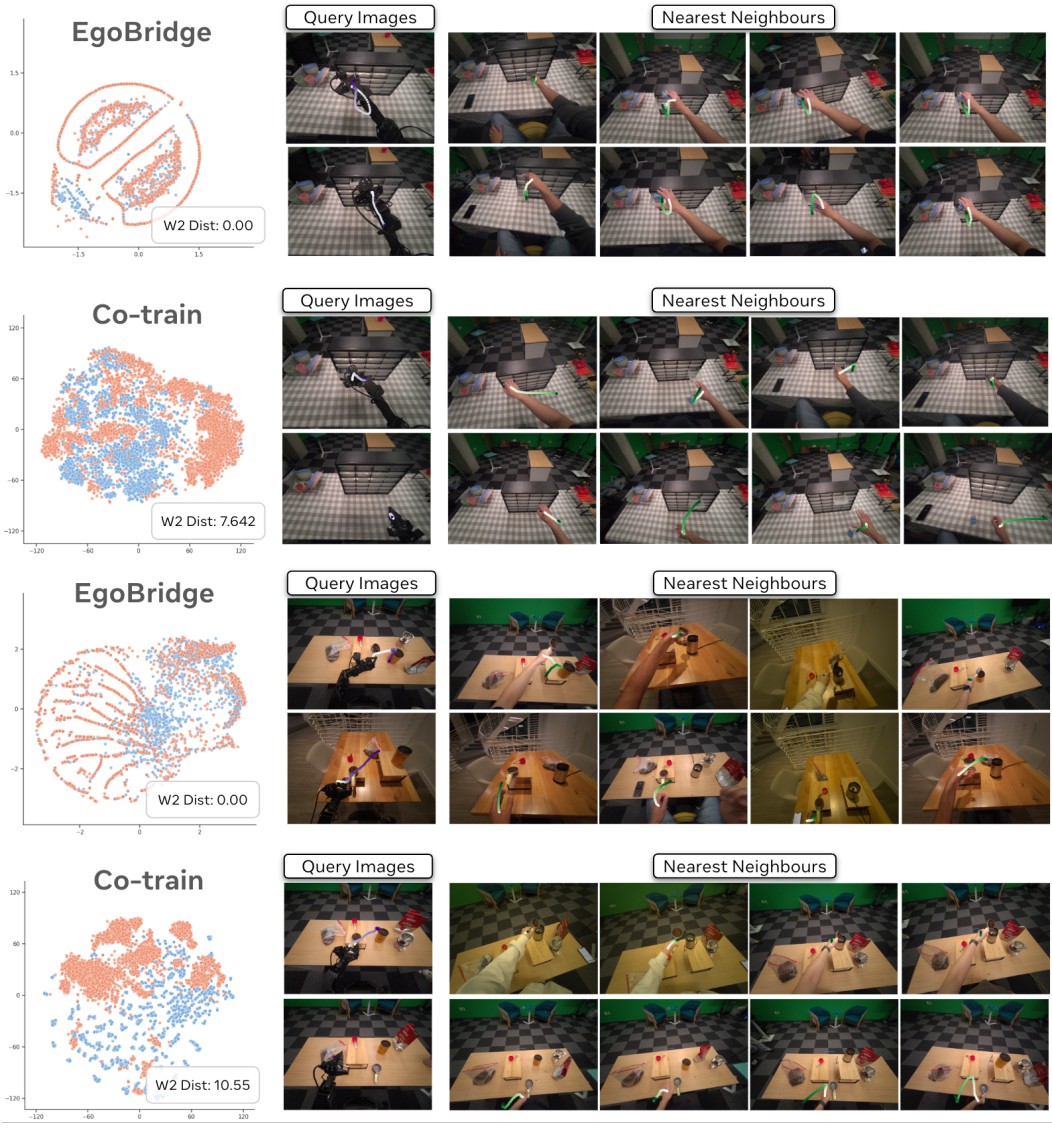

Figure 10: Qualitative visualization of the learned latent space using T-SNE and K-Nearest Neighbors.

## K    Latent Space Visualisation

We visualize the learned latent feature outputs of $f_\phi$ for the EgoBridge and Co-train models using T-SNE for the Drawer and Scoop Coffee tasks. To evaluate EgoBridge's ability to achieve joint distribution alignment, we use K-Nearest Neighbors to identify the closest learned human feature representations to a give robot feature representation. We plot the ground truth actions associated with those observations onto the input images and visualize the nearest neighbors. In addition to an overall lower Wasserstein-2 distance, which indicates global alignment, EgoBridge also aligns observations that correspond to similar behaviors in the latent space. This is visualized in Fig 10.

