# OpenReview forum: "EgoBridge: Domain Adaptation for Generalizable Imitation from Egocentric Human Data"
_NeurIPS.cc/2025/Conference — NeurIPS 2025 poster_

### Official Review · Reviewer_Eu5H · 2025-06-21

**Clarity:** 3
**Significance:** 3
**Originality:** 3
**Rating:** 4
**Confidence:** 3

**Summary:**

This paper explores methods to bridge the distribution gap between teleoperated robot data and human demonstration data, which is crucial due to differences in visual appearance, sensor modalities, and kinematics which impede effective knowledge transfer. By addressing these challenges, the study demonstrates how human demonstration data can be effectively utilized to enhance robotic learning.

**Questions:**

1. If I understand correctly, the authors use DTW to search for the most similar trajectory between the human and robot trajectories in Section 4.2. In soft supervision, the most similar trajectory should have a low transport cost. However, each trajectory consists of many steps. How are the steps aligned between the trajectories? Do the trajectories have the same length? Then, we directly align step 1 in the human trajectory to step 1 in the robot trajectory?
2. How are the trajectories divided into different steps? Human experience data might be continuous, while teleoperated data and human experience data may have different step structures to complete a task. How are these differences handled?
3. If OT is considered a type of trajectory selection method, then we align the representations between the selected teleoperated and human data. Can we directly align trajectory i with i*(j)?
4. Teleoperated data is certainly more challenging to collect. In the experiments, is the teleoperated data smaller than the human data?

**Ethical Concerns:**

["NO or VERY MINOR ethics concerns only"]

**Final Justification:**

After reading the rebuttal, my concerns have been addressed.

**Limitations:**

See Question

**Quality:**

3

**Strengths And Weaknesses:**

This paper is well-written and well-organized. The performance demonstrated is impressive, and the use of OT appears justified based on the experimental results presented. However, I think in this method, the human data and robot data should be collected from the same task. Although the experiments show the method's applicability to unseen tasks, these tasks are notably similar to those used during training. I am concerned about whether this method would perform effectively if a large amount of human data and only a minimal amount of robot data were collected on a completely new task.

---

> ### Author Rebuttal · Authors · 2025-07-31
>
> We find it encouraging that the reviewer finds that our paper is well-written and well-organized and that our experimental results are impressive.
>
> ***“[Would] this method would perform effectively if a large amount of human data and only a minimal amount of robot data were collected on a completely new task.”***
>
> We assume that sufficient human and robot data are required for domain alignment. The scope of our work to show that in single task settings we can resolve the domain gap between human and robot for
> Superior transfer for in-domain tasks over other human data augmented baselines
> Generalization to new appearances (objects and scenes)
> Generalization to new behaviours and motions that are only seen in human data
> We do agree that multi-task learning may allow one to collect less robot data on a completely new task, however the study of this is beyond the scope of this work. Our vision is that existing efforts to collect broad and diverse robot data can be augmented by an even larger distribution of human data of similar tasks across different scenes since human data with wearable devices can be collected with ease using EgoBridge.
>
> ***“How are the trajectories divided into different steps? Human experience data might be continuous, while teleoperated data and human experience data may have different step structures to complete a task. How are these differences handled?”***
>
> We thank the reviewer for this question. To clarify, we do not perform alignment at an episodic level. Instead, as described in L164-167 we align individual samples of (observation, action chunk) pairs from the human and robot dataset. An action chunk is a finite sequence of atomic actions. In our case, as mentioned in L234-235 in the paper, it is a sequence of future cartesian end-effector poses. Therefore, given a sampled mini-batch of human and robot (observation, action chunk) pairs, we identify the most similar samples with DTW and use that as soft supervision for OT to align latent embeddings computed from the samples in a differentiable manner.
>
> The reviewer asks ***“In soft supervision, the most similar trajectory should have a low transport cost. However, each trajectory consists of many steps. How are the steps aligned between the trajectories? Do the trajectories have the same length? Then, we directly align step 1 in the human trajectory to step 1 in the robot trajectory?”***
>
> The human and robot trajectories have the same length of 100. DTW captures spatial similarity between two trajectories while allowing for differences in execution speed. Each timestamp in the first trajectory is mapped to a corresponding (but potentially different) timestamp in the second and the distance is aggregated across timestamps to produce a cumulative loss. This is described in L185-192 of the paper. Qualitatively, this heuristic cumulative loss works well in finding similar trajectories as seen in the Appendix Section I.
>
> ***“If OT is considered a type of trajectory selection method, then we align the representations between the selected teleoperated and human data. Can we directly align trajectory $i$ with $i^{\ast}(j)$?”***
>
> OT is not a trajectory selection method in our method. Instead, it is a differentiable training objective which encourages the feature encoder to align the latent space of human and robot data (L171-176). This results in a coupling feature distribution which permits transfer of knowledge. The reviewer astutely observes that it may be possible to directly align the positive pairs through deterministic mapping. This is precisely an alternate deterministic formulation of Optimal Transport Plan, the Monge map. We choose to use the probabilistic “soft” version of OT (Kantarovich) [27] as in a mini-batch setting, it is robust to pairing / label noise, and works better with behavior cloning since it retains the marginal distributions of human and robot data distributions. This design decision has also been inherited from related work in joint domain adaptation that leverage OT [1, 2, 3].
>
> ***“Teleoperated data is certainly more challenging to collect. In the experiments, is the teleoperated data smaller than the human data?”***
>
> Yes, collecting teleoperated data is much more challenging to collect and in the experiments the human data is much larger than the teleoperated data. This is summarized in the Appendix Table 4, where, for the Drawer task, human data is 2.5x the amount of robot data (360 demo v/s 144 demo), the Scoop Coffee task, human data is 14x the amount of robot data (720 demo v/s 50 demo) and Laundry task, human data is 2.33x the amount of robot data (700 demo v/s 300 demo).
>
> **References**
>
>     [1] Courty, N., Flamary, R., Habrard, A., & Rakotomamonjy, A. (2017). Joint distribution optimal transportation for domain adaptation. Advances in neural information processing systems, 30.
>
>     [2] Nicolas Courty, Rémi Flamary, Devis Tuia, and Alain Rakotomamonjy. Optimal transport for domain adaptation, 2016
>
>     [3] Damodaran, B. B., Kellenberger, B., Flamary, R., Tuia, D., & Courty, N. (2018). Deepjdot: Deep joint distribution optimal transport for unsupervised domain adaptation. In Proceedings of the European conference on computer vision (ECCV) (pp. 447-463).

---

### Official Review · Reviewer_3a7U · 2025-06-23

**Clarity:** 3
**Significance:** 2
**Originality:** 2
**Rating:** 4
**Confidence:** 4

**Summary:**

This paper presents EgoBridge, a domain adaptation framework for enabling robots to learn manipulation tasks from egocentric human and robot demonstration data together. The key innovation lies in using Optimal Transport (OT) to align the joint distributions of latent policy features and actions between human and robot domains, rather than just aligning marginal distributions as in conventional domain adaptation. The method employs Dynamic Time Warping (DTW) to identify behaviorally similar human-robot action pairs, which guides the OT cost function to create meaningful correspondences across embodiments. EgoBridge addresses significant domain gaps including visual appearance differences, sensor modality mismatches, and kinematic variations between humans and robots. The framework is evaluated on both simulation and real-world manipulation tasks, demonstrating substantial improvements in success rates and crucially enabling generalization to novel objects, scenes, and behaviors that are only observed in human demonstrations.

**Questions:**

- Why does the OT cost $C_{ij}$ only apply scalar reduction to the minimum DTW pairs? Have you explored alternative strategies like graduated scaling based on $A_{ij}$ ranking order?
- For the Drawer experiments, how do the following factors affect performance: (1) varying the proportion of OOD drawers beyond 1/4, (2) different spatial arrangements of OOD drawers (random vs. clustered), and (3) different overlap ratios between human and robot training data?
- What is the Laundry task performance under out-of-distribution scenarios such as different tables, unseen shirt types, or other environmental changes?
- Given the focus on scalable human data, what is the relationship between human data proportion and robot policy performance? Also please compare with other methods across varying human-robot data ratios?

**Ethical Concerns:**

["NO or VERY MINOR ethics concerns only"]

**Final Justification:**

After reviewing the authors' rebuttal, I raise my score to **Borderline accept** for the following reasons:

> The authors have significantly clarified their proposed methodology, and the additional experiments they provided strengthen the evidence for their method's effectiveness. The rebuttal successfully addressed several of my initial concerns and enhanced my understanding of the approach.

> However, I maintain some reservations that prevent me from giving a firm **Accept**. The scaling experiments comparing human data and robot data remain incomplete, which the authors acknowledged is due to computational equipment malfunctions that prevented training new models during the rebuttal period. While I understand these technical constraints, the absence of these critical experiments limits my confidence in the work. This evaluation is particularly important because, as the authors themselves state, the core hypothesis underlying this research direction is that human data scalability surpasses robot data scalability. Without comprehensive scaling experiments, this fundamental claim remains inadequately validated.
>
> If the paper is accepted, I strongly encourage the authors to supplement their work with the missing scaling experiments, as these would provide crucial evidence for their central thesis.

I appreciate the authors' diligent efforts in addressing the reviewers' questions despite the technical challenges they faced.

**Limitations:**

# Limitations
The authors acknowledge limitations in their paper; I provide additional concerns in the **Strengths And Weaknesses** and **Questions**.

**Paper Formatting Concerns:**

No major formatting issues.

**Quality:**

2

**Strengths And Weaknesses:**

# Strengths

- **Strong empirical results**: Achieves significant performance improvements over established baselines.
- **Generalization capabilities**: Demonstrates the ability to transfer to scenarios only seen in human data, including new objects, scenes, and behavioral patterns.
- **Well-motivated technical design**: The use of DTW for identifying behaviorally similar pairs provides a robust way to handle temporal and kinematic differences between human and robot execution.

# Weaknesses

- **Limited Embodiment Generalization and Scalability Concerns**: The approach necessitates collecting human demonstrations using specialized equipment with careful calibration to ensure action space alignment, semantically similar task execution, and comparable environmental conditions between human and robot domains. This dependency means that **when switching to a different robot embodiment**, a completely new human dataset may be needed with embodiment-specific calibration and task adaptation, which significantly reduces the claimed scalability advantages over direct robot teleoperation. The paper would benefit from evaluation across different robot embodiments (e.g., different humanoid platforms or gripper configurations) using the same human dataset to demonstrate true cross-embodiment generalization.

- **Presentation Issues**: Figure 3 contains apparent errors in the evaluation case labels. The middle image in the "Evaluation Cases" section appears to be mislabeled - it should likely read "BG Color + Mirrored T" if following the authors' described experimental setup.

---

> ### Author Rebuttal · Authors · 2025-07-31
>
> We thank the reviewer for their review. We are glad that they acknowledge our strong empirical results and think that our technical design is well motivated.
>
> ***“[EgoBridge] necessitates collecting human demonstrations using specialized equipment with careful calibration to ensure action space alignment, semantically similar task execution, and comparable environmental conditions between human and robot domains.”***
>
> In regards to “using specialized equipment with careful calibration”, XR / smart wearables are growing ubiquitous. Our work simply requires some form of relative device localization at minimum. Many research devices such as Project Aria, and products such as Quest, Vision Pro with this capability are readily available. There are also many open source algorithms such as MANO and HAMER that provide 3D hand pose estimation. Large scale datasets with similar devices such as Nymeria, HOT3D, EgoExo4D, HOI4D, EgoDex etc. also leverage similar devices to collect human experience data.
>
> Using camera frame to align action spaces is a very common design choice made in prior work [1, 4, 5]. In addition hand-eye calibration to obtain the transformation between base and camera frame is a very mature and common optimization based computer vision technique which can be applied trivially on any robot embodiment and dataset.
>
> ***“when switching to a different robot embodiment, a completely new human dataset may be needed with embodiment-specific calibration and task adaptation.”***
>
> We respectfully disagree with the comment. To clarify, the human data we collect is general and **does not require any robot-specific setup or calibration.** As stated in L214-224, the data consists of first-person RGB images and proprioceptive hand tracking poses obtained directly from the Project Aria glasses. None of these assume a specific robot embodiment for datasets construction in the co-training setting. Task adaptation is also not required since we do not constrain how the human data is collected. Hence, our human data can be directly applied to co-train with data from any robot embodiment.
>
> ***“comparable environmental conditions between human and robot domains”***
>
> As stated in L289-291 and shown in Figure 4 of the paper, we collect human data on an entirely different table, lighting conditions and background for the observation generalization result where EgoBridge generalizes to a completely new environment seen only in human data.
>
> While we do agree the paper would benefit from “evaluation across different robot embodiments”, we do not have access to collect data and perform evaluation on another robot within the time constraints of the rebuttal.
>
> We thank the reviewer for pointing out presentation issues, which we will address in the next revision of the paper. Specifically we will replace the bottom right image of Figure 3 with the bottom center image. We will also replace the mislabelled bottom center case with purple background without a mirrored T to match the described evaluation cases.
>
> ***“Why does the OT cost only apply scalar reduction to the minimum DTW pairs?”***
>
> We thank the reviewer for their insightful question. We tried graduated scaling based on $A_{ij}$ by directly adding the $A_{ij}$ distance to the OT cost following prior work [6]. However, we found that this was unstable in training. We hypothesize that this is because of the large label shift between human and robot domains which makes the ordering of $A_{ij}$ noisy. Therefore we proposed a conceptually simple formulation that we also found to be stable during training empirically.
>
> ***“For the Drawer experiments, how do the following factors affect performance: (1) varying the proportion of OOD drawers beyond 1/4, (2) different spatial arrangements of OOD drawers (random vs. clustered), and (3) different overlap ratios between human and robot training data?”***
>
> From empirical observation and prior work that studies spatial generalization [2, 3], random drawer arrangements would allow the policies to interpolate over the trajectories, especially when using a variational model like Flow matching. Hence we deliberately designed a challenging setting where the test region is held out in robot data to show a clear impact of human data. As shown in both DemoGen [2], this setting is strictly harder than a random configuration. Due to time constraints of the rebuttal, we are not able to collect additional data for the specific requirement and perform training and evaluation.
>
> ***“What is the Laundry task performance under out-of-distribution scenarios such as different tables, unseen shirt types, or other environmental changes?”***
>
> To address this question, we provide an additional observation generalization experiment for the Laundry task. Here, in addition to the robot demonstrations, we add 30 minutes of human data on a **new table with white surface** unseen in robot data (we used a wooden table with brown color in robot data). We evaluate the co-trained baseline and EgoBridge on the **new table**. We perform 20 rollouts across 5 randomly sampled positions on the workspace. We calculate success rate and sub-task points as described in L292-297 of the paper. The results (success rate and points) are summarized in the table below.
>
> | Model     | Success Rate (%) | Points |
> |-----------|------------------|--------|
> | Co-train  | 25               | 25     |
> | **EgoBridge** | **50**               | **43**     |
>
> In addition, we perform an additional experiment, where we test a more challenging case of combinatorially generalizing to a **new shirt color** on the **white surface** where the new shirt is seen on the original table in the human data and excluded from robot data entirely. We use an identical evaluation protocol all the previous experiment and report the results in the table below.
>
> | Model     | Success Rate (%) | Points |
> |-----------|------------------|--------|
> | Co-train  | 5               | 7     |
> | **EgoBridge** | **30**               | **35**     |
>
> ***“Given the focus on scalable human data, what is the relationship between human data proportion and robot policy performance? Also please compare with other methods across varying human-robot data ratios?”***
>
> Prior work (EgoMimic [1]) with a similar co-training setup show a favorable scaling law for human data as compared to teleoperated robot data. We anticipate our method to show a similar or superior scaling law as we build on top of EgoMimic and show favorable performance across all evaluated tasks compared to it. We would like to support this with empirical results, however we are unable to perform this evaluation within the time constraints of the rebuttal.
>
> **References**
>
>     [1] Kareer, S., Patel, D., Punamiya, R., Mathur, P., Cheng, S., Wang, C., ... & Xu, D. (2024). Egomimic: Scaling imitation learning via egocentric video. arXiv preprint arXiv:2410.24221.
>
>     [2] Xue, Z., Deng, S., Chen, Z., Wang, Y., Yuan, Z., & Xu, H. (2025). Demogen: Synthetic demonstration generation for data-efficient visuomotor policy learning. arXiv preprint arXiv:2502.16932.
>
>     [3] Saxena, V., Bronars, M., Arachchige, N. R., Wang, K., Shin, W. C., Nasiriany, S., ... & Xu, D. (2025). What matters in learning from large-scale datasets for robot manipulation. arXiv preprint arXiv:2506.13536.
>
>     [4] Bahl, S., Mendonca, R., Chen, L., Jain, U., & Pathak, D. (2023). Affordances from human videos as a versatile representation for robotics. In Proceedings of the IEEE/CVF Conference on Computer Vision and Pattern Recognition (pp. 13778-13790).
>
>     [5] Wen, C., Lin, X., So, J., Chen, K., Dou, Q., Gao, Y., & Abbeel, P. (2023). Any-point trajectory modeling for policy learning. arXiv preprint arXiv:2401.00025.
>
>     [6] Damodaran, B. B., Kellenberger, B., Flamary, R., Tuia, D., & Courty, N. (2018). Deepjdot: Deep joint distribution optimal transport for unsupervised domain adaptation. In Proceedings of the European conference on computer vision (ECCV) (pp. 447-463).

---

> > ### Comment · Reviewer_3a7U · 2025-08-02
> >
> > Thank you for addressing my concerns. I would like to clarify some details, as I still have the following concerns:
> >
> > ***
> >
> > **Regarding Cross-Embodiment Data Transfer**
> >
> > The manipulation research community is gradually shifting focus from grippers to dexterous hands, which relates to the cross-robot embodiment challenges I mentioned. Since your current experiments utilize grippers, I wonder whether the datasets and approach could be directly applied to scenarios involving dexterous hands.
> >
> > I would find it reasonable if you could demonstrate that your human dataset's actions are derived through post-processing from hand motions (for instance, if the SE(3) actions are generated based on hand position representations). This would make sense given that human hands represent the most sophisticated embodiment we currently know for dexterous manipulation.
> >
> > **Regarding Scalability**
> >
> > Since scalability forms a key foundation of your methodology for utilizing human data, I believe experiments examining performance variations with different human-robot data ratios would be valuable and necessary to demonstrate scalability potential. For example, to support your claim that scaling human data improves robot performance, an experiment where you systematically reduce portions of the collected human training data might provide sufficient justification without requiring pretraining on extremely large datasets. I would appreciate seeing some preliminary results in this direction, as I cannot fully assess the scalability patterns of your method based solely on its EgoMimic origins, given the substantial modifications you have made.
> >
> > **Regarding the Drawer Experiments**
> >
> > The interpolation aspect you raise is intriguing. Thus, I interpret positions that can be linearly interpolated from training data as effectively being included data points (perhaps weighted as 0.5 data points? I'm not sure). This leads me to suspect that some of your out-of-distribution (OOD) drawers might still be "interpolated" from your training data when considering grasp points:
> >
> > |||||
> > |:-:|:-:|:-:|:-:|
> > |T|T|O|O|
> > |T|T|O|O|
> > |T|T|I|O|
> > |T|T|T|T|
> > |T|T|T|T|
> > |T|T|T|T|
> >
> > Here "T" represents robot training demonstration grasping positions, "I" represents the OOD grasping position covered by the convex hull of robot training positions, and "O" represents OOD grasping positions outside this convex hull.
> >
> > This observation makes me question the experimental design somewhat, which may affect my evaluation of the experiment's effectiveness. Perhaps it would be more compelling to conduct experiments where the convex hull of grasping positions cannot cover any OOD positions, such as:
> >
> > |||||
> > |:-:|:-:|:-:|:-:|
> > |T|T|T|O|
> > |T|T|T|O|
> > |T|T|T|O|
> > |T|T|T|O|
> > |T|T|T|O|
> > |T|T|T|O|
> >
> > If such experiments could be conducted and proven effective, it would better demonstrate the OOD generalization capability. Without this, I remain uncertain whether the observed capability stems from the interpolation effects you mentioned or genuinely benefits from human data. I would be grateful to see some preliminary quantitative results addressing this concern.
> >
> > ***
> >
> > Thank you again for your effort in addressing my concerns, and I would appreciate hearing your further thoughts on these comments.

---

> > > ### Author Response · Authors · 2025-08-05
> > >
> > > We thank the reviewer for acknowledging our comments and responses, and we are motivated that our answers address their concerns.
> > >
> > > ***“I would find it reasonable if you could demonstrate that your human dataset's actions are derived through post-processing from hand motions (for instance, if the SE(3) actions are generated based on hand position representations).”***
> > >
> > > As mentioned in the paper in Appendix Section E L223-323, we obtain the hand pose tracking from the Project Aria Machine Perception Services [1]. The hand pose is computed from the palm centroid keypoint of the fully-articulated 21-keypoint dextrous hand tracking. Hence, the SE(3) actions *are* directly based on dextrous hand pose representations.
> > >
> > > ***“For example, to support your claim that scaling human data improves robot performance, an experiment where you systematically reduce portions of the collected human training data might provide sufficient justification without requiring pretraining on extremely large datasets.”***
> > >
> > >
> > > We thank the reviewer for their suggestion for a supplementary experiment. Our training cluster is facing catastrophic failures, which are making it challenging for us to provide results for this experiment. We are trying our best to provide preliminary results for this experiment before the culmination of the rebuttal period.
> > >
> > > ***“This leads me to suspect that some of your out-of-distribution (OOD) drawers might still be "interpolated" from your training data when considering grasp points. This observation makes me question the experimental design somewhat, which may affect my evaluation of the experiment's effectiveness.”***
> > >
> > > We thank the reviewer for their insightful comment and the informative diagram. As suggested, we construct a new table by omitting the “I” / interpolated drawer from the computation of the OOD success rate and report the results for the Drawer task. Specifically, we use the following evaluation setting, where we exclude the convex hull (X denotes excluded). The performance of our method decreased from 33% to 30%. Most other baselines remain to have zero success rate, while ATM went from 8% to 0% in success rate. We note that while it is possible for end-to-end policies trained with sufficient data to exhibit a certain degree of spatial interpolation or even extrapolation, this amended table clearly shows the impact of human data and that EgoBridge enables better transfer from human data compared to other co-training methods, which is the key result of this work.
> > >
> > > |     |     |     |     |
> > > |-----|-----|-----|-----|
> > > | T   | T   | O   | O   |
> > > | T   | T   | O   | O   |
> > > | T   | T   | X   | O   |
> > > | T   | T   | T   | T   |
> > > | T   | T   | T   | T   |
> > > | T   | T   | T   | T   |
> > >
> > >  Method      | OOD Success Rate (%) |
> > > |-------------|----------------------|
> > > | Robot BC    | 0                    |
> > > | Co-train    | 0                    |
> > > | EgoMimic    | 0                    |
> > > | ATM         | 0                    |
> > > | Mimicplay   | 0                    |
> > > | **EgoBridge** | **30**               |
> > >
> > > **References**
> > >
> > >     [1] Meta Research, “Basics — project aria docs,” https://facebookresearch.github.io/projectaria tools/docs/data formats/mps/mps summary, 2024, accessed: September 15, 2024

---

> > > > ### Comment · Reviewer_3a7U · 2025-08-05
> > > >
> > > > I appreciate the authors' efforts in addressing my concerns despite the constraints and difficulties they faced. The alternative experiments provided are valuable, particularly given that training new models is currently not feasible for the authors. The additional clarifications have provided helpful information that addresses my questions. I will update my review accordingly based on these responses.

---

### Official Review · Reviewer_kgpd · 2025-06-28

**Clarity:** 3
**Significance:** 3
**Originality:** 3
**Rating:** 5
**Confidence:** 4

**Summary:**

The paper presents EgoBridge, a framework for co-training robot manipulation policies using both egocentric human data and robot teleoperation data. The key innovation lies in using Optimal Transport-informed cost function to explicitly align joint latent representations (observation embeddings and actions) across human and robot domains. This alignment facilitates effective cross-embodiment policy learning and generalization. Extensive evaluations in simulation and three real-world tasks demonstrate the co-training approach significantly improves policy success rate, particularly in settings where objects, scenes, or behaviors appear only in human data.

**Questions:**

See the strengths and weaknesses section.
Additional questions:
- Currently the human data is not very diverse. If human data collection can be scaled up and includes more diverse scenarios, would the policy be able to generalize to scenarios neither seen in human or robot data?
- Have you compared to contrastive or VLM-based alignment methods as an alternative to OT?
- How expensive is the OT computation per training step? Does it impact training scalability in larger datasets?
- How critical are the wrist camera inputs in robot data compared to the egocentric view? Would EgoBridge still be effective without them?

**Ethical Concerns:**

["NO or VERY MINOR ethics concerns only"]

**Final Justification:**

The authors clarified the computation cost and limitations of the approach during rebuttal. All my concerns are addressed and I lean towards accept.

**Limitations:**

Yes

**Quality:**

3

**Strengths And Weaknesses:**

Strengths
- The approach is sound and well-motivated. By aligning joint (latent policy feature, action) distributions using differentiable OT, the approach preserves action-relevant information—addressing limitations of prior domain adaptation strategies that align marginal distributions only. Experiments are thorough and ablations are well-executed.
- EgoBridge shows impressive results, it can improve in-domain task performance, generalize to objects and scenes only seen in human data, and motions only seen in human data.
- The paper is well-written and the structure is clear. The method is described in detail, including the formulation of the OT cost and its integration into the training objective. Figures (e.g., t-SNE plots) are informative and intuitive to understand.

Weaknesses
- The reliance and generalizability on DTW for action similarity may become impractical in multi-task or more diverse data regimes, as acknowledged in the paper.
- While EgoBridge outperforms baselines and ablations, the overall task success rates are still pretty low. What is preventing the tasks to get 80-90% success rates? More failure case analysis would be helpful.
- Although EgoBridge enables some extent of generalization to new scenarios only seen in human data (e.g., unseen objects, scenes, motions), the “new” scenarios are not very different from robot training distribution. Can the policy generalize to objects of different shapes and sizes (not only appearance), that require different contact modes? Can the policy generalize to new strategies or modes only seen in human data?

Minor paper writing issues
- In Fig 2, the arrows are not aligned.
- In Fig 4, the legends are confusing. The blue "robot" legend does not show up anywhere in the images. In (a), there are two shades of purple outlines, it is unclear what is the difference between them.

---

> ### Author Rebuttal · Authors · 2025-07-31
>
> We thank the reviewer for their positive review. We are encouraged that they find our results impressive and find the paper to be well written.
>
> ***“The reliance and generalizability on DTW for action similarity may become impractical in multi-task or more diverse data regimes”***
>
> We thank the reviewer for the insightful comment. We acknowledge that this is a valid concern especially in multi-task settings where DTW-based action pairings may lead to incorrect or degenerate pairings, as discussed in L365-368 of the main text. However, the generality of our joint domain adaptation formulation allows for the inclusion of additional heuristics and labels. For instance, language embedding distances (as explored in adjacent works[1]) and dense language annotations of actions are becoming the norm in large scale imitation learning and would allow us to extend EgoBridge to more complex and multi-task settings in future works.
>
> ***“What is preventing the tasks to get 80-90% success rates?”***
>
> We deliberately chose challenging long-horizon tasks with considerable reset variation as these are challenging settings that require a large amount of teleoperated demonstrations and hence can benefit the most from human data. The laundry task involved manipulation of a deformable object, the coffee task involves fine-grained grasping of a scoop with considerable reset variation and the drawer task requires very precise placement of the object. We could further improve the performance with additional robot data, but herein lies the unique benefit of EgoBridge of leveraging human data in resource-constrained settings, where we can get 70+% success in challenging tasks such as Laundry with only 300 demonstrations across a large workspace.
>
> ***“Currently the human data is not very diverse. If human data collection can be scaled up and includes more diverse scenarios, would the policy be able to generalize to scenarios neither seen in human or robot data?”***
>
> Our work focuses in establishing joint domain adaptation as a method for imitation learning to align data from robot and human domains to enable transfer of knowledge from the source (human) to target (robot) domain. As such, we chose to focus our experiments on validating and studying the transfer of knowledge and whether the policy can generalize to scenarios only shown in human data. While achieving zero-shot generalization to new scenarios is an important problem for the field, we consider it to be an orthogonal direction that would require a dedicated, full-fledged research study.
>
> ***“Can the policy generalize to objects of different shapes and sizes (not only appearance), that require different contact modes?”***
>
> Grasp primitives are not captured in the action space (cartesian). Hence given the  large embodiment gap (parallel jaw v/s dextrous) in grasp, we hypothesize that grasping new object shapes will be challenging.
>
> ***Can the policy generalize to new strategies or modes only seen in human data?”***
>
> We thank the reviewer for this question. We acknowledge that generalization to new modes is a challenging task. We show this in EgoBridge, through the drawer task where the robot needs to place the object in target drawers where there is no spatial robot data, hence achieving motion generalization through human data. Future work will study other modes and behaviours we can transfer from human data.
>
> ***“How expensive is the OT computation per training step? Does it impact training scalability in larger datasets?”***
>
> Thanks for the clarification question. We use the geomloss [2] library which is a kernel optimized sinkhorn process that runs in parallel on GPU. The run-time of this algorithm is $O(N^2)$ where N is the batch size. In practice OT computation is a fraction of the time compared to the model forward pass. To empirically validate this claim, we use a single L40s GPU, with batch size 32 and torch.cuda.synchronize() to profile over 10 batches.
>
> *Average forward pass time (human + robot) :* 0.344373s \
> *Average OT Loss time :* 0.005504s
>
> Which makes the $T^{\ast}$ computation roughly 1.6% of the forward pass time and hence it doesn’t impact training scalability with larger batch sizes and datasets.
>
> ***“Have you compared to contrastive or VLM-based alignment methods as an alternative to OT?”***
>
> Yes, we have experimented with contrastive-based alignment methods as an alternative. We empirically observed that when naively applied, contrastive learning results in unstable training. Specifically, we utilized the same DTW action supervision to form positive pairs and used the InfoNCE [3] loss on policy latents. We hypothesize that this instability is a result of contrastive learning not retaining the marginal distributions of the human and robot, which is crucial for behavior cloning. Additional research is required to correctly adapt contrastive learning for joint domain adaptation.
>
> ***“How critical are the wrist camera inputs in robot data compared to the egocentric view? Would EgoBridge still be effective without them?”***
>
> We thank the reviewer for the insightful comments. We had a similar hypothesis that removing wrist camera may improve alignment, but we empirically found that the wrist camera is critical to manipulation steps that require precision, such as grasping and alignment, as echoed by many other research [4, 5] . Hence we designed the architecture to handle heterogeneous sensor modalities from the human and robot data.
>
> We thank the reviewer for identifying presentation issues, which we will address in the next revision of the paper. Specifically, we will align the arrows in Figure 2, replace the purple legend with Human + Robot, remove the robot legend and make the colors consistent in Figure 4.
>
> **References**
>
>     [1] Yu, A., Foote, A., Mooney, R., & Martín-Martín, R. (2024). Natural language can help bridge the sim2real gap. arXiv preprint arXiv:2405.10020.
>
>     [2] Feydy, J., Séjourné, T., Vialard, F. X., Amari, S. I., Trouvé, A., & Peyré, G. (2019, April). Interpolating between optimal transport and mmd using sinkhorn divergences. In The 22nd international conference on artificial intelligence and statistics (pp. 2681-2690). PMLR.
>
>     [3] Oord, A. V. D., Li, Y., & Vinyals, O. (2018). Representation learning with contrastive predictive coding. arXiv preprint arXiv:1807.03748.
>
>     [4] Liu, Y., Romeres, D., Jha, D. K., & Nikovski, D. (2020). Understanding multi-modal perception using behavioral cloning for peg-in-a-hole insertion tasks. arXiv preprint arXiv:2007.11646.
>
>     [5] Mandlekar, A., Xu, D., Wong, J., Nasiriany, S., Wang, C., Kulkarni, R., ... & Martín-Martín, R. (2021). What matters in learning from offline human demonstrations for robot manipulation. arXiv preprint arXiv:2108.03298.

---

> > ### Comment · Reviewer_kgpd · 2025-08-05
> >
> > Thanks to the authors for the detailed and thoughtful response. My concerns have been addressed and I will maintain my score.

---

### Official Review · Reviewer_nDAS · 2025-06-29

**Clarity:** 4
**Significance:** 3
**Originality:** 3
**Rating:** 5
**Confidence:** 4

**Summary:**

The paper proposes EgoBridge, an approach to improving co-training on teleop data and human data via domain adaptation. Specifically, the authors identify that the current naïve robot-human data co-training has the problem of misaligned latent representations of two types of data, and the authors propose EgoBridge that explicitly uses Optimal Transport to align the latent representation of two kinds of data. The authors show that EgoBridge can help improve the performance of co-training and enable the policy to complete tasks that have been seen in robot data and only seen in human data.

**Questions:**

I have listed my main concerns in Weakness section. Apart from that, I also have an additional question on using action similarity as soft supervision. What if we directly use it as a hard supervision instead of soft supervision? For example, for each human demo, we find the robot demo that has the highest similary to it, and then directly minimizes the distance between their latent representation. This is equivalent to directly setting T* to be 1 for the most similar pair and 0 other wise. In this way, we don’t need to update T* during training and it might make training more stable. I wonder how would the performance be if we do this?

**Ethical Concerns:**

["NO or VERY MINOR ethics concerns only"]

**Final Justification:**

The rebuttal from the authors has resolved all my concerns, and I will remain my rating of Accept, considering the paper should have good impact on robot-human data co-training in robotics.

**Limitations:**

yes

**Quality:**

3

**Strengths And Weaknesses:**

Strength:

The proposed approach is well motivated, theoretically solid, and shows great performance. The idea of using action similarity as soft supervision to construct the cost function in OT is very clever. The experiments are structured, well thought, and shows promising results.

Weakness:

1.	One main concern I have about the paper is using action similarity as soft supervision in the cost function of OT. The intuition here is when two trajectories have similar actions, they are more likely to be the same task and in the same environment setting, and thus their latent representation of the observations should also be similar. However, this assumption can be broken sometimes. For example, consider two tasks of picking a cup and opening a drawer. If the cup and the drawer are at similar position, then most part of the trajectories is very similar (the part of reaching to the object) and maybe only the last part of the trajectories (how to manipulate the object) is different. In this case, the similarity between the actions of two trajectories can be high, but we should have similar observation representation of these two trajectories because they are different tasks and have different objects. What if, instead of using action similarity as soft supervision, we use similarity of visual observation (e.g., similarity between their DINOv2 features), or similarity between language instructions, or some kind of combination of these three? Will it make the soft supervision more reliable?
2.	It seems the authors haven’t provided many details about the efficiency of training OT. One main concern is that, since the OT loss is dependent on the optimal transport plan T* and T* is the result of a optimization problem dependent on the distribution of latent representation of human and robot data, then 1) is it costly to solve the optimization problem and find T*? 2) when we are updating the model during training, the distribution of the latent representation is changing and thus the T* might also change. How to we track the changing optimality of T*? Do we continuously update T* during training and how? Is it expensive to continuously update it during training?

---

> ### Author Rebuttal · Authors · 2025-07-31
>
> We thank the reviewer for their positive evaluation of our proposed method. We are encouraged that they found our method clever and results promising!
>
> ***"For example, consider two tasks of picking a cup and opening a drawer. If the cup and the drawer are at similar position, then most part of the trajectories is very similar (the part of reaching to the object) and maybe only the last part of the trajectories (how to manipulate the object) is different."***
>
> We thank the reviewer for the insightful comment. We acknowledge that this is a valid concern especially in multi-task settings where DTW-based action pairings may lead to incorrect or degenerate pairings, as discussed in L365-368 of the main text. However, the generality of our joint domain adaptation formulation allows for the inclusion of additional heuristics and labels. For instance, language embedding distances (as explored in adjacent works[1]) and dense language annotations of actions are becoming the norm in large scale imitation learning and would allow us to extend EgoBridge to more complex and multi-task settings in future works.
>
> ***"is it costly to solve the optimization problem and find $T^{\ast}$?***
>
> Thanks for the clarification question. We use the geomloss [5] library which is a kernel optimized sinkhorn process that runs in parallel on GPU. The run-time of this algorithm is $O(N^2)$ where N is the batch size. In practice finding $T^{\ast}$ is a fraction of the time compared to the model forward pass. To empirically validate this claim, we use a single L40s GPU, with batch size 32 and torch.cuda.synchronize() to profile over 10 batches.
>
> *Average forward pass time (human + robot) :* 0.344373s \
> *Average OT Loss time :* 0.005504s
>
> Which makes the $T^{\ast}$ computation ~1.6% of the forward pass time.
>
> ***“when we are updating the model during training, the distribution of the latent representation is changing and thus the $T^{\ast}$ might also change. How to we track the changing optimality of $T^{\ast}$?”***
>
> We thank the reviewer for the insightful comment. $T^{\ast}$ computed on the policy latent is indeed evolving with the training process. $T^{\ast}$ is computed for each batch at each training step. Since we don’t use $T^{\ast}$ as a push forward but instead as a differentiable objective guiding the direction of moving probability mass from the two distributions, it acts as the supervision to the encoder to move the latent representations such that it reaches a cost-minimizing $T^{\ast}$ for all sampled batches across the dataset.
>
> Adjacent to this, the reviewer asks ***“What if we directly use it as a hard supervision instead of soft supervision? For example, for each human demo, we find the robot demo that has the highest similarity to it, and then directly minimizes the distance between their latent representation?”***
>
> This is precisely an alternate deterministic formulation of Optimal Transport Plan, the Monge map. We choose to use the probabilistic “soft” version of OT (Kantarovich) [27] as in a mini-batch setting, it is robust to pairing / label noise, and works better with behavior cloning since it retains the marginal distributions of human and robot data distributions. This design decision has also been inherited from related work in joint domain adaptation that leverage OT [2, 3, 4].
>
> **References**
>
>     [1] Yu, A., Foote, A., Mooney, R., & Martín-Martín, R. (2024). Natural language can help bridge the sim2real gap. arXiv preprint arXiv:2405.10020.
>
>     [2] Courty, N., Flamary, R., Habrard, A., & Rakotomamonjy, A. (2017). Joint distribution optimal transportation for domain adaptation. Advances in neural information processing systems, 30.
>
>     [3] Nicolas Courty, Rémi Flamary, Devis Tuia, and Alain Rakotomamonjy. Optimal transport for domain adaptation, 2016
>
>     [4] Damodaran, B. B., Kellenberger, B., Flamary, R., Tuia, D., & Courty, N. (2018). Deepjdot: Deep joint distribution optimal transport for unsupervised domain adaptation. In Proceedings of the European conference on computer vision (ECCV) (pp. 447-463).
>
>     [5] Feydy, J., Séjourné, T., Vialard, F. X., Amari, S. I., Trouvé, A., & Peyré, G. (2019, April). Interpolating between optimal transport and mmd using sinkhorn divergences. In The 22nd international conference on artificial intelligence and statistics (pp. 2681-2690). PMLR.

---

> > ### Comment · Reviewer_nDAS · 2025-08-04
> > **Thank you for the response**
> >
> > I would like to thank the authors for the thoughtful response. All of my concerns have been resolved. I will keep my rating as Accept.

---

### Note · Authors · 2025-08-13

We sincerely thank all the reviewers for their thoughtful feedback, positive evaluations, and constructive suggestions on our work, EgoBridge. We are encouraged by the positive reception of our paper, with reviewers finding our method **clever**, our results **impressive**, and our technical design **well-motivated**. Across the board, reviewers highlighted the potential of our **novel joint domain adaptation** formulation for leveraging large-scale human data to benefit robot learning. The insightful questions raised during the review process have been invaluable in helping us clarify key aspects of our work and strengthen our paper.

We addressed several feedback from the reviewers and provided detailed clarifications regarding:
- *The scalability and computational cost of the proposed OT loss*: Here, we demonstrated empirically that the OT computation is a negligible fraction (∼1.6%) of the model's forward pass time and permits model scalability
- *The robustness and flexibility of our joint domain adaptation formulation,* which permits the addition of additional heuristics such as language embeddings to handle complex and multi-task scenarios in future work.
- *The generalization capabilities of our approach,* where we showcased new experimental results for the Laundry task that demonstrate successful generalization to new tables and unseen shirt colors, which validates our method’s ability to generalize to out-of-distribution scenarios
- *Data collection requirements*: Here, we clarified that our human data collection is general and can be applied to any robot embodiment, which emphasizes our method’s applicability to future robotics research.
- *Our design choices,* where we emphasized the advantages of using the “soft” Kantorovich formulation of Optimal Transport over a deterministic Monge map and alternative methods like contrastive learning.

We also committed to incorporating the suggested presentation improvements into our final paper revision to enhance its clarity and readability further.

We are pleased that the reviewers unanimously agree on the significance and generality of our contributions and have found the clarifications and additional experiments provided during the rebuttal process to be satisfactory. We look forward to refining the presentation and extending the evaluation in future work.

---

### Decision · Program_Chairs · 2025-09-17

**Decision:**

Accept (poster)

**Comment:**

The paper proposes EgoBridge to improve co-training on human and robot data via domain adaptation. It explicitly uses Optimal Transport to align the latent representations of the two types of data.

The initial major concerns from the reviewers include:
- Using action similarity only as soft supervision in the cost function of OT (nDAS)
- Lack of details on the efficiency of training OT (nDAS,kgpd )
- Generalization and scalability (kgpd, 3a7U)

The rebuttal adequately addressed these concerns, and the final ratings of the paper are 2 Accepts and 2 Borderline Accepts. The AC agrees with the reviewers' assessment and recommends acceptance. The authors should revise the paper accordingly in the final version.